# RAINPROOF: An umbrella to shield text generators from Out-of-Distribution data

**Maxime Darrin**[*]
ILLS[†]
MILA - Quebec AI Institute
McGill University
Paris-Saclay University

**Pablo Piantanida**
ILLS[†]
MILA - Quebec AI Institute
CNRS, CentraleSupélec
Paris-Saclay University

**Pierre Colombo**
MICS[‡]
CentraleSupélec
Paris-Saclay University
Equall, Paris

## Abstract

Implementing effective control mechanisms to ensure the proper functioning and security of deployed NLP models, from translation to chatbots, is essential. A key ingredient to ensure safe system behaviour is Out-Of-Distribution (OOD) detection, which aims to detect whether an input sample is statistically far from the training distribution. Although OOD detection is a widely covered topic in classification tasks, most methods rely on hidden features output by the encoder. In this work, we focus on leveraging soft-probabilities in a black-box framework, *i.e.* we can access the soft-predictions but not the internal states of the model. Our contributions include: (i) RAINPROOF a Relative informAItioN Projection OOD detection framework; and (ii) a more operational evaluation setting for OOD detection. Surprisingly, we find that OOD detection is not necessarily aligned with task-specific measures. The OOD detector may filter out samples well processed by the model and keep samples that are not, leading to weaker performance. Our results show that RAINPROOF provides OOD detection methods more aligned with task-specific performance metrics than traditional OOD detectors.

## 1 Introduction

Significant progress has been made in Natural Language Generation (NLG) in recent years with the development of powerful generic (*e.g.,* GPT (Radford et al., 2018; Brown et al., 2020; Bahrini et al., 2023), LLAMA (Touvron et al., 2023) and its variants) and task-specific (*e.g.,* Grover (Zellers et al., 2019), Pegasus (Zhang et al., 2020) and Dialog-GPT (Zhang et al., 2019b)) text generators. They power machine translation (MT) systems or chatbots that are exposed to the public, and their reliability is a prerequisite for adoption. Text generators are trained in the context of a so-called closed

---

[*]maxime.darrin@mila.quebec
[†]ILLS - International Laboratory on Learning Systems
[‡]Mathématiques et Informatique Centralesupelec

world (Fei and Liu, 2016), where training and test data are assumed to be drawn *i.i.d.* from a single distribution, known as the in-distribution. However, when deployed, these models operate in an open world (Parmar et al., 2021; Zhou, 2022) where the *i.i.d.* assumption is often violated. Changes in data distribution are detrimental and induce a drop in performance. It is necessary to develop tools to protect models from harmful distribution shifts as it is a clearly unresolved practical problem (Arora et al., 2021). For example, a trained translation model is not expected to be reliable when presented with another language (*e.g.* a Spanish model exposed to Catalan, or a Dutch model exposed to Afrikaans) or unexpected technical language (*e.g.,* a colloquial translation model exposed to rare technical terms from the medical field). They also tend to be released behind API(OpenAI, 2023) ruling out many usual features-based OOD detection methods.

Most work on Out-Of-Distribution (OOD) detection focus on classification, leaving OOD detection in (conditional) text generation settings mainly unexplored, even though it is among the most exposed applications. Existing solutions fall into two categories. The first one called *training-aware methods* (Zhu et al., 2022; Vernekar et al., 2019a,b), modifies the classifier training by exposing the neural network to OOD samples during training. The second one, called *plug-in methods* aims to distinguish regular samples in the in-distribution (IN) from OOD samples based on the model's behaviour on a new input. Plug-in methods include Maximum Softmax Probabilities (MSP) (Hendrycks and Gimpel, 2016) or Energy (Liu et al., 2020) or feature-based anomaly detectors that compute a per-class anomaly score (Ming et al., 2022; Ryu et al., 2017; Huang et al., 2020; Ren et al., 2021a). Although plug-in methods from classification settings seem attractive, their adaptation to text generation tasks is more involved. While text generation can be seen as a sequence of classification problems, i.e.,

chosing the next token at each step, the number of possible tokens is two orders of magnitude higher than usual classification setups.

In this work, we aim to develop new tools to build more reliable text generators which can be used in practical systems. To do so, we work under 4 constraints: (i) We do not assume we can access OOD samples; (ii) We suppose we are in a *black-box scenario*: we do not assume we have access to the internal states of the model but only to the soft probability distributions it outputs; (iii) The detectors should be easy enough to use on top of any existing model to ensure adaptability; (iv) Not only should OOD detectors be able to filter OOD samples, but they also are expected to improve the average performance on the end-task the model has to perform.

**Our contributions.** Our main contributions can be summarized as follows:

1. *A more operational benchmark for text generation OOD detection.* We present LOFTER the **L**anguage **O**ut o**F** dis**T**ribution p**E**rformance benchma**R**k. Existing works on OOD detection for language modelling (Arora et al., 2021) focus on (i) the English language only, (ii) the GLUE benchmark, and (iii) measure performance solely in terms of OOD detection. LOFTER introduces more realistic data shifts in the generative setting that goes beyond English: language shifts induced by closely related language pairs (*e.g.*, Spanish and Catalan or Dutch and Afrikaans (Xiao et al., 2020)[1]) and domain change (*e.g.*, medical vs news data or vs dialogs). In addition, LOFTER comes with an updated evaluation setting: detectors' performance is jointly evaluated *w.r.t* the overall system's performance on the end task.

2. *A novel detector inspired by information projection.* We present RAINPROOF: a **R**elative inform**AI**tio**N** **P**rojection **O**ut **OF** distribution detector. RAINPROOF is fully **unsupervised**. It is flexible and can be applied both when no reference samples (IN) are available (corresponding to scenario $s_0$) and when they are (corresponding to scenario $s_1$). RAINPROOF tackles $s_0$ by computing the models' predictions negentropy (Brillouin, 1953) and uses it as a measure of normality. For $s_1$, it relies upon its natural extension: the Information Projection

(Csiszar and Matus, 2003), which relies on a reference set to get a data-driven notion of normality.

3. *New insights on the operational value of OOD detectors.* Our experiments on LOFTER show that *OOD detectors may filter out samples that are well processed (i.e. well translated) by the model and keep samples that are not, leading to weaker performance.* Our results show that RAINPROOF improves performance on the end task while removing most of the OOD samples.

4. *Code and reproductibility.* We release a plug-and-play library built upon the Transformers library that implements our detectors and baselines [2] [3].

## 2 Problem Statement & Related Works

### 2.1 Notations & conditional text generation

Let us denote $\Omega$, a vocabulary of size $|\Omega|$ and $\Omega^*$ its Kleene closure[4]. We denote $\mathcal{P}(\Omega) = \left\{ \mathbf{p} \in [0,1]^{|\Omega|} : \sum_{i=1}^{|\Omega|} \mathbf{p}_i = 1 \right\}$ the set of probability distributions defined over $\Omega$. Let $\mathcal{D}_{train}$ be the training set, composed of $N \geqslant 1$ i.i.d. samples $\{(\mathbf{x}^i, \mathbf{y}^i)\}_{i=1}^N \in (\mathcal{X} \times \mathcal{Y})^N$ with probability law $\mathbf{p}_{XY}$. We denote $\mathbf{p}_X$ and $\mathbf{p}_Y$ the associated marginal laws of $\mathbf{p}_{XY}$. Each $\mathbf{x}^i$ is a sequence of tokens, and $x_j^i \in \Omega$ the $j$th token of the $i$th sequence. $\mathbf{x}_{\leqslant t}^i = \{x_1^i, \cdots, x_t^i\} \in \Omega^*$ denotes the prefix of length $t$. The same notations hold for $\mathbf{y}$.

**Conditional textual generation.** In conditional textual generation, the goal is to model a probability distribution $\mathbf{p}_\star(\mathbf{x}, \mathbf{y})$ over variable-length text sequences $(\mathbf{x}, \mathbf{y})$ by finding $\mathbf{p}_\theta \approx \mathbf{p}_\star(\mathbf{x}, \mathbf{y})$ for any $(\mathbf{x}, \mathbf{y})$. In this work, we assume to have access to a pretrained conditional language model $f_\theta : \mathcal{X} \times \mathcal{Y} \to \mathbf{R}^{|\Omega|}$, where the output is the (unnormalized) logits scores. $f_\theta$ parameterized $\mathbf{p}_\theta$, *i.e.*, for any $(\mathbf{x}, \mathbf{y})$, $\mathbf{p}_\theta(\mathbf{x}, \mathbf{y}) = \text{softmax}(f_\theta(\mathbf{x}, \mathbf{y})/T)$ where $T \in \mathbf{R}$ denotes the temperature. Given an input sequence $\mathbf{x}$, the pretrained language $f_\theta$ can recursively generate an output sequence $\hat{\mathbf{y}}$ by sampling $y_{t+1} \sim \mathbf{p}_\theta^T(\cdot | \mathbf{x}, \hat{\mathbf{y}}_{\leqslant t})$, for $t \in [1, |\mathbf{y}|]$. Note that $\hat{y}_0$ is the start of sentence ($< \text{SOS} >$ token). We denote by $\mathcal{S}(\mathbf{x})$, the set of normalized logits scores generated by the model when the initial input is $\mathbf{x}$ *i.e.,* $\mathcal{S}(\mathbf{x}) = \{\text{softmax}(f_\theta(\mathbf{x}, \hat{\mathbf{y}}_{\leqslant t}))\}_{t=1}^{|\hat{\mathbf{y}}|}$.

---

[1]Afrikaans is a daughter language of Dutch. The Dutch sentence: "Appelen zijn gewoonlijk groen, geel of rood" corresponds to "Appels is gewoonlik groen, geel of rooi."

[2]https://github.com/icannos/ToddBenchmark

[3]This work was performed using HPC resources from GENCI–IDRIS (Grant 2022-AD011013945).

[4]The Kleene closure corresponds to sequences of arbitrary size written with words in $\Omega$. Formally: $\Omega^* = \bigcup_{i=0}^{\infty} \Omega^i$.

Note that elements of $\mathcal{S}(\mathbf{x})$ are discrete probability distributions over $\Omega$.

## 2.2 Problem statement

In OOD detection, the goal is to find an anomaly score $a : \mathcal{X} \to \mathbf{R}_+$ that quantifies how far a sample is from the IN distribution. $\mathbf{x}$ is classified as IN or OUT according to the score $a(\mathbf{x})$. We then fix a threshold $\gamma$ and classifies the test sample IN if $a(\mathbf{x}) \leqslant \gamma$ or OOD if $a(\mathbf{x}) > \gamma$. Formally, let us denote $g(\cdot, \gamma)$ the decision function, we take:
$$g(\mathbf{x}, \gamma) = \begin{cases} 1 & \text{if } a(\mathbf{x}) > \gamma \\ 0 & \text{if } a(\mathbf{x}) \leqslant \gamma. \end{cases}$$
*Remark* 1. *In our setting, OOD examples are not available.* Tuning $\gamma$ is a complex task, and it is usually calibrated using OOD samples. In our work, we decided not to rely on OOD samples but on the available training set to fix $\gamma$ in a realistic setting. Indeed, even well-tailored datasets might contain significant shares of outliers (Meister et al., 2023). Therefore, we fix $\gamma$ so that at least $80\%$ of the IN data pass the filtering procedure. See Sec. G.3 for more details.

## 2.3 Review of existing OOD detectors

**OOD detection for classification.** Most works on OOD detection have focused on detectors for classifiers and rely either on internal representations (*features-based detectors*) or on the final soft probabilities produced by the classifier (*softmax based detectors*).

**Features-based detectors.** They leverage latent representations to derive anomaly scores. The most well-known is the Mahanalobis distance (Lee et al., 2018a; Ren et al., 2021b), but there are other methods employing Grams matrices (Sastry and Oore, 2020), Fisher Rao distance (Gomes et al., 2022) or other statistical tests (Haroush et al., 2021). These methods require access to the latent representations of the models, which does not fit the black-box scenario. *In addition, it is well known in classification that performing per-class OOD detection is key to get good performance* (Lee et al., 2018b). This per-class approach is *a priori* impossible in text generation since it would have to be done per token or by some other unknown type of classes. We argue that it is necessary to find non-class-dependent solutions, especially when it comes to the Mahalanobis distance, which relies upon the hypothesis that the data are unimodal; we study the validity of this hypothesis and show that it is not true in a generative setting in Ap. A.

**Softmax-based detectors.** These detectors rely on the soft probabilities produced by the model. The MSP (Hendrycks and Gimpel, 2017; Hein et al., 2019; Liang et al., 2018; Hsu et al., 2020) uses the probability of the mode while others take into account the entire logit distribution (*e.g.*, Energy-based scores (Liu et al., 2020)). Due to the large vocabulary size, it is unclear how these methods generalize to sequence generation tasks.

**OOD detection for text generation.** Little work has been done on OOD detection for text generation. Therefore, we will follow (Arora et al., 2021; Podolskiy et al., 2021) and rely on their baselines. We also generalize common OOD scores such as MSP or Energy by computing the average score along the sequence at each step of the text generation. We refer the reader to Sec. B.7 for more details.

**Quality estimation as OOD detection metric.** Quality Estimation Metrics are not designed to detect OOD samples but to assess the overall quality of generated samples. However, they are interesting baselines to consider, as OOD samples should lead to low-quality outputs. We will use COMET QE (Stewart et al., 2020) as a baseline to filter out low-quality results induced by OOD samples.

*Remark* 2. Note that *features-based* detectors assume white-box access to internal representations, while *softmax-based* detectors rely solely on the final output. Our work operates in a black-box framework but also includes a comparison to the Mahalanobis distance for completeness.

## 3 RAINPROOF OOD detector

### 3.1 Background

An information measure $\mathcal{I} : \mathcal{P}(\Omega) \times \mathcal{P}(\Omega) \to \mathbf{R}$ quantifies the similarity between any pair of discrete distributions $\mathbf{p}, \mathbf{q} \in \mathcal{P}(\Omega)$. Since $\Omega$ is a finite set, we will adopt the following notations $\mathbf{p} = [\mathbf{p}_1, \cdots, \mathbf{p}_{|\Omega|}]$ and $\mathbf{q} = [\mathbf{q}_1, \cdots, \mathbf{q}_{|\Omega|}]$. While there exist information distances, it is, in general, difficult to build metrics that satisfy all the properties of a distance, thus we often rely on divergences that drop the symmetry property and the triangular inequality.

In what follows, we motivate the information measures we will use in this work.

First, we rely on the *Rényi divergences* (Csiszár, 1967). Rényi divergences belong to the $f$-divergences family and are parametrized by a parameter $\alpha \in \mathbf{R}_+ - \{1\}$. They are flexible

and include well-known divergences such as the Kullback-Leiber divergence (KL) (Kullback, 1959) (when $\alpha \to 1$) or the Hellinger distance (Hellinger, 1909) (when $\alpha = 0.5$). The Rényi divergence between $\mathbf{p}$ and $\mathbf{q}$ is defined as follows:

$$D_\alpha(\mathbf{p}\|\mathbf{q}) = \frac{1}{\alpha - 1} \log \left( \sum_{i=1}^{|\Omega|} \frac{\mathbf{p}_i^\alpha}{\mathbf{q}_i^{\alpha-1}} \right). \quad (1)$$

The Rényi divergence is popular as $\alpha$ allows weighting the relative influence of the distributions' tail.

Second, we investigate the *Fisher-Rao distance* (FR). FR is a distance on the Riemannian space formed by the parametric distributions, using the Fisher information matrix as its metric. It computes the geodesic distance between two discrete distributions (Rao, 1992) and is defined as follows:

$$\text{FR}(\mathbf{p}\|\mathbf{q}) = \frac{2}{\pi} \arccos \sum_{i=1}^{|\Omega|} \sqrt{\mathbf{p}_i \times \mathbf{q}_i}. \quad (2)$$

It has recently found many applications (Picot et al., 2022; Colombo et al., 2022b).

### 3.2 `RAINPROOF` for the no-reference scenario ($\mathbb{s}_0$)

At inference time, the no-reference scenario ($\mathbb{s}_0$) does not assume the existence of a reference set of IN samples to decide whether a new input sample is OOD. Which include, for example, Softmax-based detectors such as MSP, Energy or the sequence log-likelihood[5]

Under these assumptions, our OOD detector `RAINPROOF` comprises three steps. For a given input $\mathbf{x}$ with generated sentence $\hat{\mathbf{y}}$:

1. We first use $f_\theta$ to extract the step-by-step sequence of soft distributions $\mathcal{S}(\mathbf{x})$.

2. We then compute an anomaly score ($a_\mathcal{I}(\mathbf{x})$) by averaging a step-by-step score provided by $\mathcal{I}$. This step-by-step score is obtained by measuring the similarity between a reference distribution $\mathbf{u} \in \mathcal{P}(\Omega)$ and one element of $\mathcal{S}(\mathbf{x})$. Formally,

$$a_\mathcal{I}(\mathbf{x}) = \frac{1}{|\mathcal{S}(\mathbf{x})|} \sum_{\mathbf{p} \in \mathcal{S}(\mathbf{x})} \mathcal{I}(\mathbf{p}\|\mathbf{u}), \quad (3)$$

where $|\mathcal{S}(\mathbf{x})| = |\hat{\mathbf{y}}|$.

---

[5]The detector based on the log-likelihood of the sequence is defined as $a_L(\mathbf{x}) = -\frac{1}{|\hat{\mathbf{y}}|} \sum_{t=0}^{|\hat{\mathbf{y}}|-1} \log \mathbf{p}_\theta(\hat{\mathbf{y}}_{t+1}|\mathbf{x}, \hat{\mathbf{y}}_{\leqslant t})$.

3. The last step consists of thresholding the previous anomaly score $a_\mathcal{I}(\mathbf{x})$. If $a_\mathcal{I}(\mathbf{x})$ is over a given threshold $\gamma$, we classify $\mathbf{x}$ as an OOD example.

**Interpretation of Eq. 3.** $a_\mathcal{I}(\mathbf{x})$ measures the average dissimilarity of the probability distribution of the next token to normality (as defined by $\mathbf{u}$). $a_\mathcal{I}(\mathbf{x})$ also corresponds to the token average uncertainty of the model $f_\theta$ to generate $\hat{\mathbf{y}}$ when the input is $\mathbf{x}$. The intuition behind Eq. 3 is that the distributions produced by $f_\theta$, when exposed to an OOD sample, should be far from normality and thus have a high score.

**Choice of $\mathbf{u}$ and $\mathcal{I}$.** The uncertainty definition of Eq. 3 depends on the choice of both the reference distribution $\mathbf{u}$ and the information measure $\mathcal{I}$. A natural choice for $\mathbf{u}$ is the uniform distribution, *i.e.,* $\mathbf{u} = [\frac{1}{|\Omega|}, \cdots, \frac{1}{|\Omega|}]$ which we will use in this work. It is worth pointing out that $\mathcal{I}(\cdot\|\mathbf{u})$ yields the negentropy of a distribution (Brillouin, 1953). Other possible choices for $\mathbf{u}$ include one hot or tf-idf distribution (Colombo et al., 2022b). For $\mathcal{I}$, we rely on the Rényi divergence to obtain $a_{\mathcal{D}_\alpha}$ and the Fisher-Rao distance to obtain $a_{\text{FR}}$.

### 3.3 `RAINPROOF` for the reference scenario ($\mathbb{s}_1$)

In the reference scenario ($\mathbb{s}_1$), we assume that one has access to a reference set of **IN samples** $\mathcal{R} = \{\mathbf{x}^i : (\mathbf{x}^i, \mathbf{y}^i) \in \mathcal{D}_{train}\}_{i=1}^{|\mathcal{R}|}$ where $|\mathcal{R}|$ is the size of the reference set. For example, the Mahalanobis distance works under this assumption. One of the weaknesses of Eq. 3 is that it imposes an ad-hoc choice when using $\mathbf{u}$ (the uniform distribution). In $\mathbb{s}_1$, we can leverage $\mathcal{R}$, to obtain a *data-driven notion normality*.

Under $\mathbb{s}_1$, our OOD detector `RAINPROOF` follows these four steps:

1. (*Offline*) For each $\mathbf{x}^i \in \mathcal{R}$, we generate $\hat{\mathbf{y}}^i$ and the associated sequence of probability distributions ($\mathcal{S}(\mathbf{x}^i)$). Overall we thus generate $\sum_{\mathbf{x} \in \mathcal{R}} |\hat{\mathbf{y}}^i|$ probability distributions which could explode for long sequences[6]. To overcome this limitation, we rely on the bag of distributions of each sequence (Colombo et al., 2022b). We form the set of these

---

[6]It is also worth pointing out that projecting at each timestep would require a per-step reference set in addition to the computational time required to compute the projections, therefore we decided to aggregate the probability distributions over the sequence.

bags of distributions:

$$\bar{\mathcal{S}}^* = \bigcup_{\mathbf{x}^i \in \mathcal{R}} \left\{ \frac{1}{|\mathcal{S}(\mathbf{x}^i)|} \sum_{\mathbf{p} \in \mathcal{S}(\mathbf{x}^i)} \mathbf{p} \right\}. \qquad (4)$$

2. (*Online*) For a given input $\mathbf{x}$ with generated sentence $\hat{\mathbf{y}}$, we compute its bag of distributions representation:

$$\bar{\mathbf{p}}(\mathbf{x}) = \frac{1}{|\mathcal{S}(\mathbf{x})|} \sum_{\mathbf{p} \in \mathcal{S}(\mathbf{x})} \mathbf{p}. \qquad (5)$$

3. (*Online*) For $\mathbf{x}$, we then compute an anomaly score $a_{\mathcal{I}}^{\star}(\mathbf{x})$ by projecting $\bar{\mathbf{p}}(\mathbf{x})$ on the set $\bar{\mathcal{S}}^*$. Formally, $a_{\mathcal{I}}^{\star}(\mathbf{x})$ is defined as:

$$a_{\mathcal{I}}^{\star}(\mathbf{x}) = \min_{\mathbf{p} \in \bar{\mathcal{S}}^{\star}} \mathcal{I}(\mathbf{p} \| \bar{\mathbf{p}}(\mathbf{x})). \qquad (6)$$

We denote $\mathbf{p}^{\star}(\mathbf{x}) = \arg \min_{\mathbf{p} \in \bar{\mathcal{S}}^*} \mathcal{I}(\mathbf{p} \| \bar{\mathbf{p}}(\mathbf{x}))$.

4. The last step consists of thresholding the previous anomaly score $a_{\mathcal{I}}(\mathbf{x})$. If $a_{\mathcal{I}}(\mathbf{x})$ is over a given threshold $\gamma$, we classify $\mathbf{x}$ as an OOD example.

**Interpretation of Eq. 6.** $a_{\mathcal{I}}(\mathbf{x})$ relies on a Generalized Information Projection (Kullback, 1954; Csiszár, 1975, 1984)[7] which measures the similarity between $\bar{\mathbf{p}}(\mathbf{x})$ and the set $\bar{\mathcal{S}}^*$. Note that the closest element of $\bar{\mathcal{S}}^*$ in the sens of $\mathcal{I}$ can give insights on the decision of the detector. It allows interpreting the decision of the detector as we will see in Tab. 6.

**Choice of $\mathcal{I}$.** Similarly to Sec. 3.2, we will rely on the Rényi divergence to define $a_{\mathcal{R}_\alpha}^{\star}(\mathbf{x})$ and the Fisher-Rao distance $a_{\mathrm{FR}}^{\star}(\mathbf{x})$.

## 4 Results on LOFTER

### 4.1 LOFTER: Language Out oF disTribution pErformance benchmaRk

**LOFTER for NMT.** We consider a realistic setting involving both topic and language shifts. *Language shifts* correspond to exposing a model trained for a given language to another which is either linguistically close (*e.g.*, Afrikaans for a system trained on Dutch) or missing in the training data (as it is the case for german in BLOOM (Scao et al., 2022)). It is an interesting setting because the differences between languages might not be obvious but still

cause a significant drop in performance. For linguistically close languages, we selected closely related language pairs such as Catalan-Spanish, Portuguese-Spanish and Afrikaans-Dutch) coming from the Tatoeba dataset (Tiedemann, 2012) (see Tab. 8). *Domain shifts* can involve technical or rare terms or specific sentence constructions, which can affect the model's performance. We simulated such shifts from Tatoeba MT using news, law (EuroParl dataset), and medical texts (EMEA).

**LOFTER for dialogs.** For conversational agents, we focused on a scenario where a goal-oriented agent, designed to handle a specific type of conversation (*e.g.*, customer conversations, daily dialogue), is exposed to an unexpected conversation. In this case, it is crucial to interrupt the agent so it does not damage the user's trust with misplaced responses. We rely on the Multi WOZ dataset (Zang et al., 2020), a human-to-human dataset collected in the Wizard-of-Oz set-up (Kelley, 1984), for IN distribution data and its associated fine-tuned model. We simulated shifts using dialogue datasets from various sources, which are part of the SILICONE benchmark (Chapuis et al., 2020). Specifically, we use a goal-oriented dataset (*i.e.*, Switchboard Dialog Act Corpus (SwDA) (Stolcke et al., 2000)), a multi-party meetings dataset (*i.e.*, MRDA (Shriberg et al., 2004) and Multimodal EmotionLines Dataset MELD (Poria et al., 2018)), daily communication dialogs (*i.e.*, DailyDialog DyDA (Li et al., 2017)), and scripted scenarii (*i.e.*, IEMOCAP (Tripathi et al., 2018)). We refer the curious reader to Sec. B.5 for more details on each dataset.

**Model Choices.** We evaluated our methods on open-source and freely available language bilingual models (the Helsinki suite (Tiedemann and Thottingal, 2020)), on a BLOOM-based instructions model BLOOMZ (Muennighoff et al., 2022) (for which German is OOD). For dialogue tasks, we relied on the Dialog GPT (Zhang et al., 2019b) model finetuned on Multi WOZ, which acts as IN distribution. We consider the Helsinki models as they are used in production for lightweight applications. Additionally, they are specialized for a specific language pair and released with their associated training set, making them ideal candidates to study the impact of OOD in a controlled setting.[8]

**Metrics.** To evaluate the performance on the OOD task, we report the *Area Under the Receiver Oper-*

---

[7]The minimization problem of Eq. 6 finds numerous connections in the theory of large deviation (Sanov, 1958) or in statistical physics (Jaynes, 1957).

[8]Please note that for the likelihood detector, the translation model is additionally fine-tuned on the development set, ensuring a strong baseline.

Table 1: **Summary of the OOD detection performance** of our detectors (Ours) compared to commonly used strong baselines (Bas.). We report the best detector for each scenario in bold and underline the best overall. The ↓ indicates that for this score, the lower, the better; otherwise, the higher, the better.[9]

| | | | Language shifts | | Domain shifts | | Dialog shifts | |
|---|---|---|---|---|---|---|---|---|
| | | | AUROC | FPR ↓ | AUROC | FPR ↓ | AUROC | FPR ↓ |
| $\mathfrak{s}_0$ | Ours | $a_{D_\alpha}$ | **0.95** | **0.25** | **0.85** | **0.62** | **0.79** | **0.64** |
| | | $a_{FR}$ | 0.93 | 0.28 | 0.81 | 0.67 | 0.76 | 0.68 |
| | Bas. | $a_E$ | 0.89 | 0.44 | 0.79 | 0.78 | 0.65 | 0.76 |
| | | $a_{MSP}$ | 0.87 | 0.44 | 0.79 | 0.77 | 0.66 | 0.72 |
| | | $a_L$ | 0.78 | 0.79 | 0.73 | 0.88 | 0.65 | 0.95 |
| | | $a_{CQE}$ | 0.71 | 0.57 | 0.73 | 0.88 | X | X |
| $\mathfrak{s}_1$ | Ours | $a_{D_\alpha^*}$ | 0.88 | 0.34 | **0.86** | **0.50** | **0.86** | **0.52** |
| | | $a_{FR^*}$ | 0.88 | 0.35 | 0.81 | 0.69 | 0.76 | 0.75 |
| | Bas. | $a_M$ | **0.92** | **0.26** | 0.78 | 0.59 | 0.84 | 0.55 |

Table 2: **Correlation between OOD scores and translation metrics** BLEU, BERT-S and COMET.

| | | | BERT-S | | | BLEU | | | COMET | | |
|---|---|---|---|---|---|---|---|---|---|---|---|
| | | Score | ALL | IN | OUT | ALL | IN | OUT | ALL | IN | OUT |
| $\mathfrak{s}_0$ | Ours | $a_{D_\alpha}$ | -0.48 | -0.33 | -0.53 | -0.35 | -0.27 | -0.38 | -0.33 | -0.23 | -0.22 |
| | | $a_{FR}$ | -0.45 | -0.32 | -0.45 | -0.35 | -0.28 | -0.37 | -0.36 | -0.28 | -0.26 |
| | Bas. | $a_E$ | -0.08 | 0.02 | -0.44 | 0.04 | 0.06 | -0.13 | 0.21 | 0.25 | 0.13 |
| | | $a_L$ | 0.23 | 0.25 | -0.23 | 0.30 | 0.28 | 0.06 | 0.44 | 0.44 | 0.23 |
| | | $a_{MSP}$ | -0.13 | -0.01 | -0.43 | 0.01 | 0.04 | -0.14 | 0.15 | 0.21 | 0.12 |
| | | $a_{CQE}$ | 0.09 | 0.11 | 0.08 | 0.11 | 0.10 | 0.21 | 0.45 | 0.45 | 0.74 |
| $\mathfrak{s}_1$ | Ours | $a_{D_\alpha^*}$ | -0.33 | -0.20 | -0.65 | -0.21 | -0.16 | -0.35 | -0.11 | -0.03 | -0.11 |
| | | $a_{FR^*}$ | -0.31 | -0.20 | -0.64 | -0.20 | -0.15 | -0.37 | -0.11 | -0.04 | -0.15 |
| | Bas. | $a_M$ | -0.21 | -0.12 | -0.29 | -0.09 | -0.05 | -0.05 | -0.06 | -0.01 | 0.01 |

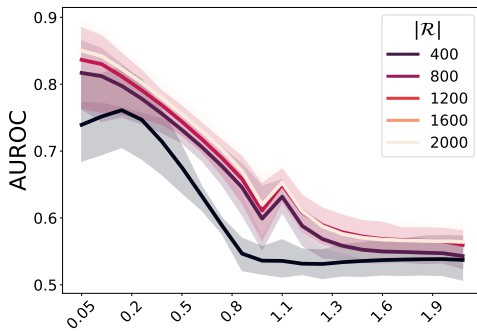

Figure 1: **Ablation study on** RAINPROOF for $\alpha$ and reference set size ($|\mathcal{R}|$) for dialogue shift detection. Smaller $\alpha$ emphasizes the tail of the distribution, while $\alpha = 0$ counts common non-zero elements.

*ating Characteristic* AUROC and the *False positive rate* FPR ↓. These methods have been widely employed in previous research on out-of-distribution (OOD) detection. An exhaustive description of the metrics can be found in Sec. 4.1.

## 4.2 Experiments in MT

**Results on language shifts (Tab. 1).** We find that our no-reference methods ($a_{D_\alpha}$ and $a_{FR}$) achieve better performance than common no-reference baselines but also outperform the reference-based baseline. In particular, $a_{D_\alpha}$, by achieving an AUROC of 0.95 and FPR ↓ of 0.25, outperforms all considered methods. Moreover, while no-reference baselines only capture up to $45\%$ of the OOD samples on average, ours detect up to $55\%$. In addition, COMET QE, a quality estimation tool, performs poorly in pure OOD detection, suggesting that while OOD detection and quality estimation can be related, they are still different problems.

**Results on domain shifts (Tab. 1).** We evaluate the OOD detection performance of RAINPROOF on domain shifts in Spanish (SPA) and German (DEU) with technical, medical data and parliamentary data. For $\mathfrak{s}_0$, we observe that $a_{D_\alpha}$ and $a_{FR}$ outperform the strongest baselines (*i.e.*, Energy, MSP and se-

quence likelihood) by several AUROC points. Interestingly enough, even our no-reference detectors outperform the reference-based baseline (*i.e.*, $a_M$, a deeper study of this phenomenon is presented in Ap. A). While $a_{D_\alpha}$ achieves similar AUROC performance to its information projection counterpart $a_{D_\alpha^*}$, the latter achieve better FPR ↓. Once again, the COMET QE metric does not yield competitive performance for OOD detection.

## 4.3 Experiments in dialogue generation

**Results on Dialogue shifts (Tab. 1).** Dialogue shifts are understandably more difficult to detect, as shown in our experiments, as they are smaller than language shifts. Our no-reference detectors do not outperform the Mahalanobis baseline and achieve only 0.79 in AUROC. The best baseline is the Mahalanobis distance and achieves better performance on dialogue tasks than on NMT domain shifts, reaching an AUROC of 0.84. However, *our reference-based detector based on the Rényi information projection secures better* AUROC *(*0.86*) and better* FPR ↓ *(*0.52*)*. Even if our detectors achieve decent results on this task, it is clear that dialogue shifts will require further work and investigation (see Ap. F), especially in the wake of LLMs.

## 4.4 Ablations Study

Fig. 1 shows that RAINPROOF offers a crucial flexibility by utilizing the Rényi divergence with adjustable parameter alpha. RAINPROOF's detectors show improvement when considering the tail of the distributions. Notably, lower values of $\alpha$ (close to 0) yield better results with the Rényi Information projection $a_{D_\alpha^*}$. This finding suggests that the tail of the distributions used in text generation contains contextual information and insights about the processed texts. These results are consistent

Table 3: **Impact of OOD detectors on BLEU** for IN data only, OOD data and the combination of both ALL. We report average BLEU (Abs.), BLEU gains (G.s) compared to $f_\theta$ only, and removed subset share (R.Sh.). $\gamma$ set to remove 20% of IN dataset.

| | | IN Abs. | IN G.s | IN R.Sh | OOD Abs. | OOD G.s | OOD R.Sh | ALL Abs. | ALL G.s | ALL R.Sh |
|---|---|---|---|---|---|---|---|---|---|---|
| | ✗ | 53.7 | +0.0 | 0.0% | 30.8 | +0.0 | 0.0% | 48.1 | +0.0 | 0.0% |
| $\mathfrak{s}_0$ Ours | $a_{FR}$ | 57.1 | +3.4 | 17.6% | 34.7 | +3.9 | 46.1% | 54.9 | +6.7 | 24.9% |
| | $a_{D_\alpha}$ | 56.1 | +2.4 | 19.9% | 39.9 | +9.1 | 62.1% | 54.8 | +6.7 | 31.8% |
| Bas. | $a_E$ | 56.7 | +3.0 | 20.0% | 31.9 | +1.1 | 32.0% | 52.1 | +3.9 | 22.7% |
| | $a_L$ | 58.1 | +4.4 | 19.0% | 35.0 | +4.2 | 44.4% | 55.3 | +7.2 | 25.5% |
| | $a_{MSP}$ | 52.2 | -1.5 | 18.4% | 26.7 | -4.1 | 38.5% | 46.5 | -1.7 | 25.8% |
| | $a_{CQE}$ | 54.7 | +1.0 | 20.0% | 32.6 | +1.8 | 20.9% | 49.2 | +1.1 | 20.5% |
| $\mathfrak{s}_1$ Ours | $a_{D_\alpha^*}$ | 53.9 | +0.2 | 19.2% | 31.1 | +0.3 | 60.1% | 51.0 | +2.8 | 32.0% |
| | $a_{FR^*}$ | 53.9 | +0.2 | 19.0% | 31.1 | +0.3 | 59.9% | 51.0 | +2.8 | 31.8% |
| Bas. | $a_M$ | 53.7 | -0.0 | 20.0% | 31.9 | +1.1 | 61.4% | 50.4 | +2.2 | 33.5% |

Table 4: **Computation time (in sec).** Off. (Onl.) stands for offline (resp. online) time.

| Score | Off. | Onl. |
|---|---|---|
| $a_{D_\alpha}$ | ✗ | $2.10^{-3}$ s |
| $a_{MSP}$ | ✗ | $1.10^{-4}$ s |
| $a_M$ | 40s | $3.10^{-3}$ s |
| $a_{D_\alpha^*}$ | ✗ | $9.10^{-2}$ s |

with recent research in automatic text generation evaluation (Colombo et al., 2022b). Interestingly, increasing the size of the reference set beyond 1.2k has minimal influence. We provide an additional study of the impact of the temperature and parameter $\alpha$ for the different OOD scores in Ap. E.

## 5 A More Practical Evaluation

Following previous work, we measure the performance of the detectors on the OOD detection task based on AUROC and FPR $\downarrow$. However, this evaluation framework neglects the impact of the detector on the overall system's performance and the downstream task it performs. We identify three main evaluation criteria that are important in practice: (i) *execution time*, (ii) *overall system performance* in terms of the quality of the generated answers, and (iii) *interpretability of the decision*. Our study is conducted on NMT because due to the existence of relevant and widely adopted metrics for assessing the quality of a generated sentence (*i.e.*, BLEU (Papineni et al., 2002) and BERT-S (Zhang et al., 2019a) and COMET (Stewart et al., 2020)).

### 5.1 Execution time

**Runtime and memory costs.** We report in Tab. 4 the runtime of all methods. Detectors for $\mathfrak{s}_0$ are faster than the ones for $\mathfrak{s}_1$. Unlike detectors using references, no-reference detectors do not require additional memory. They can be set up easily in a plug&play manner at virtually no costs.

### 5.2 Effects of Filtering on Translation Quality

In this experiment, we investigate the impact of OOD filtering from the perspectives of quality estimation and selective generation.

**Global performance.** In Tab. 3 and Tab. 5, we report the global performance of the system ($f_\theta$) with and without OOD detectors on IN samples, OOD samples, and all samples (ALL). In most cases, adding detectors increases the average quality of the returned answers on all three subsets but with varying efficacy. $a_{MSP}$ is a notable exception, and we provide a specific correlation analysis later. While the reference-based detectors tend to remove more OOD samples, the no-reference detectors demonstrate better performance regarding the remaining sentences' average BLEU. Thus, *OOD detector evaluation should consider the final task performance.* Overall, it is worth noting that directly adapting classical OOD detection methods (*e.g.*, MSP or Energy) to the sequence generation problem leads to poor results in terms of performance gains (*i.e.*, as measured by BLEU or BERT-S). $a_{D_\alpha}$ removes up to 62% of OOD samples (whereas the likelihood only removes 45%) and maintains or improves the average performance of the system on the end task. *In other words, $a_{D_\alpha}$ provides the best combination of OOD detection performance and system performance improvements.*

**Threshold free analysis.** In Tab. 2, we report the correlations between OOD scores and quality metrics on each data subset (IN and OUT distribution, and ALL combined). For the OOD detector to improve or maintain performance on the end task, its score must correlate with performance metrics similarly for each subset. We notice that it is not the case for the likelihood or $a_{MSP}$. The highest likelihood on IN data corresponds to higher quality answers. Still, the opposite is true for OOD samples, meaning using the likelihood to remove OOD samples tends to remove OOD samples that are well handled by the model. By contrast, RAINPROOF scores correlate well and in the same way on both IN and OUT, allowing them to remove OOD samples while improving performance.

### 5.3 Towards an interpretable decision

An important dimension of fostering adoption is the ability to verify the decision taken by the automatic system. RAINPROOF offers a step in this direction when used with references: for each input sample, RAINPROOF finds the closest sample

Table 5: Detailed impacts on NMT performance results per tasks (Domain- or Language-shifts) of the different detectors. We present results on the different parts of the data: IN data, OOD data and the combination of both, ALL. For each, we report the absolute average BLEU (Abs.), the average gains in BLEU (G.s.) compared to a setting without OOD filtering ($f_\theta$ only) and the share of the subset removed by the detector (R.Sh.). We provide more detailed results on each dataset in Ap. G. In addition, we performed this study using different thresholds see Sec. G.3

| | | | Domain shifts | | | | | | | | Language shifts | | | | | | | | |
| | | | IN | | | OOD | | | ALL | | | IN | | | OOD | | | ALL | | |
| | | | Abs. | G. | Rh. | Abs. | G. | Rh. | Abs. | G. | Rh. | Abs. | G. | Rh. | Abs. | G. | Rh. | Abs. | G. | Rh. |
|---|---|---|---|---|---|---|---|---|---|---|---|---|---|---|---|---|---|---|---|---|
| | | ✗ | 46.9 | +0.0 | 0.0% | 43.3 | +0.0 | 0.0% | 46.2 | +0.0 | 0.0% | 60.5 | +0.0 | 0.0% | 18.3 | +0.0 | 0.0% | 50.1 | +0.0 | 0.0% |
| $S_0$ | Ours | $a_{FR}$ | 49.9 | +3.0 | 18.2% | 46.8 | +3.5 | 23.0% | 50.0 | +3.8 | 19.9% | 64.3 | +3.8 | 17.0% | 22.6 | +4.3 | 69.3% | 59.7 | +9.6 | 29.9% |
| | | $a_{D_\alpha}$ | 49.0 | +2.1 | 20.0% | 46.2 | +3.0 | 40.9% | 48.0 | +1.9 | 27.8% | 63.2 | +2.7 | 19.8% | 33.6 | +15.3 | 83.2% | 61.6 | +11.5 | 35.8% |
| | Bas. | $a_E$ | 49.4 | +2.6 | 20.0% | 45.5 | +2.3 | 18.0% | 48.9 | +2.8 | 19.7% | 63.9 | +3.4 | 19.9% | 18.4 | +0.0 | 46.0% | 55.2 | +5.1 | 25.8% |
| | | $a_L$ | 50.6 | +3.8 | 19.0% | 47.5 | +4.3 | 24.2% | 50.9 | +4.7 | 21.1% | 65.6 | +5.1 | 19.0% | 22.4 | +4.1 | 64.6% | 59.7 | +9.6 | 30.0% |
| | | $a_{MSP}$ | 45.6 | -1.3 | 18.8% | 33.4 | -9.9 | 45.7% | 42.1 | -4.1 | 29.6% | 58.9 | -1.7 | 18.1% | 20.0 | +1.7 | 31.3% | 50.8 | +0.7 | 22.1% |
| | | $a_{CQE}$ | 48.0 | +1.2 | 20.0% | 44.2 | +0.9 | 17.1% | 46.8 | +0.6 | 20.2% | 61.3 | +0.8 | 20.0% | 21.1 | +2.8 | 24.8% | 51.6 | +1.5 | 20.9% |
| $S_1$ | Ours | $a_{D_\alpha}^*$ | 46.9 | +0.0 | 19.0% | 37.4 | -5.9 | 63.1% | 46.3 | +0.1 | 35.0% | 60.9 | +0.4 | 19.5% | 24.7 | +6.4 | 57.0% | 55.6 | +5.5 | 29.0% |
| | | $a_{FR}^*$ | 46.9 | +0.0 | 18.7% | 37.4 | -5.9 | 63.0% | 46.3 | +0.1 | 34.8% | 60.9 | +0.4 | 19.2% | 24.7 | +6.4 | 56.7% | 55.6 | +5.5 | 28.7% |
| | Bas. | $a_M$ | 46.7 | -0.1 | 20.0% | 43.1 | -0.1 | 62.7% | 45.4 | -0.7 | 36.5% | 60.6 | +0.1 | 20.0% | 20.6 | +2.3 | 60.1% | 55.3 | +5.2 | 30.4% |

Table 6: **Interpretability Analysis** OOD source (S.), their ground-truth (GD.), their generation (Gen.) and projections onto the reference set.

| S. | Ahir a la nit vàrem treballar fins a les deu. | |
|---|---|---|
| Gd. | Last night we worked until 10 p.m. | |
| Gen. | Ahir a la nit vàrem treballar fins a les deu. | BLEU 3.75 |
| $\mathbf{p}^\star(\mathbf{x})$ | Dar gato por liebre. | **Score** 1.23 |
| S | Austràlia no és Àustria. | |
| Gd | Australia isn't Austria. | |
| Gen. | Austràlia is not Austria. | BLEU 21.86 |
| $\mathbf{p}^\star(\mathbf{x})$ | La vida no es fácil. | **Score** 0.82 |

Table 7: **OOD detection on BLOOMZ** using German as an OOD language for the LLM.

| | | | AUROC | FPR ↓ |
|---|---|---|---|---|
| $S_0$ | Ours | $a_{FR}$ | 0.50 | 1.00 |
| | | $a_{D_\alpha}$ | 0.58 | 0.92 |
| | Bas. | $a_{MSP}$ | **0.59** | **0.90** |
| | | $a_E$ | 0.51 | 0.97 |
| | | $a_L$ | 0.54 | 0.90 |
| $S_1$ | Ours | $a_{D_\alpha}^*$ | **0.71** | **0.80** |
| | | $a_{FR}^*$ | 0.70 | 0.82 |
| | Bas. | $a_M$ | 0.66 | 0.89 |

(in the sense of the Information Projection) in the reference set to take its decision. Tab. 6 present examples of OOD samples along with their translation scores, projection scores, and projection on the reference set. Qualitative analysis shows that, in general, sentences close to the reference set and whose projection has a close meaning are better handled by $f_\theta$. Therefore, one can visually interpret the prediction of RAINPROOF and validate it.

## 6 RAINPROOF on LLM for NMT

As an alternative to NMT models, we can study the performance of instruction finetuned LLM on translation tasks. However, it is important to note that while LLMs are trained on enormous amounts of data, they still miss many languages. Typically, they are trained on around 100 languages (Conneau et al., 2019), this falls far short of the existing 7000 languages. In our test-bed experiments, we decided to rely on BLOOM models, which have not been specifically trained on German (DEU) data. Therefore, We can use German samples to simulate OOD detection in an instruction-following translation setting, specifically relying on BLOOMZ (Muennighoff et al., 2022). We prompt the model to trans-

late Tatoeba dataset samples into English, focusing on languages known to be within the distribution for BLOOMZ while attempting to separate the German samples from them. From Tab. 7, we observed that our no-reference methods perform comparably to the $a_{MSP}$ baseline, but are outperformed by the Mahalanobis distance in this scenario. *However, the information projection methods demonstrate substantial improvements over all the baselines.*

## 7 Conclusion

This work introduces a detection framework called RAINPROOF and a new benchmark called LOFTER for black-box OOD detection on text generator. We adopt an operational perspective by not only considering OOD performance but also task-specific metrics: despite the good results obtained in pure OOD detection, *OOD filtering can harm the performance of the final system, as it is the case for $a_{MSP}$ or $a_M$.* We found that RAINPROOF succeed in removing OOD while inducing significant gains in translation performance both on OOD samples and in general. In conclusion, this work paves the way for developing text-generation OOD detectors and calls for a global evaluation when benchmarking future OOD detectors.

## 8   Limitations

While this work does not bear significant ethical or impact hazards, it is worth pointing out that it is not a perfect, absolutely safe solution against OOD distribution samples. Preventing the processing of OOD samples is an important part of ensuring ML algorithms' safety and robustness but it cannot guarantee total safety nor avoid all OOD samples. In this work, we approach the problem of OOD detection from a performance standpoint: we argue that OOD detectors should increase performance metrics since they should remove risky samples. However, no one can give such guarantees, and the outputs of ML models should always be taken with caution, whatever safety measures or filters are in place. Additionally, we showed that our methods worked in a specific setting of language shifts or topic shifts, mainly on translation tasks. While our methods performed well for small language shifts (shifts induced by linguistically close languages) and showed promising results on detecting topic shifts, the latter task remains particularly hard. Further work should explore different types of distribution shifts in other newer settings such as different types of instructions or problems given to instruction-finetuned models.

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

## A Examining the Limitations of Mahalanobis-Based OOD Detector for Text Generation

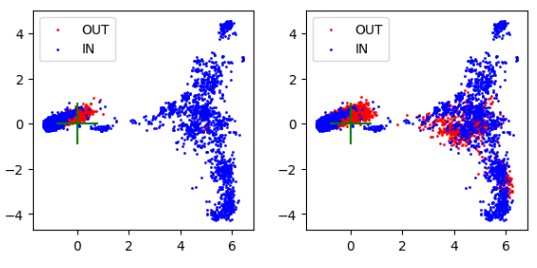

(a) *deu* date on *fra* model.  (b) *spa* date on *fra* model.

Figure 2: **PCA reduction of encoder's hidden features** for IN and OUT distribution samples, with Mahalanobis distance mean (green cross). The plot reveals the multimodal nature of the distributions.

The main drawback of the Mahalanobis distance is assuming a single-mode distribution. In text classification, this is mitigated by fitting one Mahalanobis scorer per class. However, in text generation, this assumption is flawed as there are multiple modes as illustrated in Fig. 2). PCA of Fig. 2 illustrate a failure case of the Mahalanobis distance in the case of OOD detection.

## B Experimental setting

In this section, we dive into the details and definitions of our experimental setting. First, we present our OOD detection performance metrics (Sec. B.1), then we provide a couple samples for one of the small language shifts (Sec. B.4). We also discuss the choices of pre-trained model (Sec. B.6) and how we adapted common OOD detectors to the text generation case (Sec. B.7).

### B.1 Additionnal details on metrics

OOD Detection is usually an unbalanced binary classification problem where the class of interest is OUT. Let us denote $Z$ the random variable corresponding to actually being out of distribution. We can assess the performance of our OOD detectors focusing on the **False alarm rate** and on the **True detection rate**. The **False alarm rate** or False positive rate (FPR) is the proportion of samples misclassified as OUT. For a score threshold $\gamma$, we have FPR $= \Pr\left(a(\mathbf{x}) > \gamma \,|\, Z = 0\right)$. The **True detection rate** or True positive rate (TPR) is the proportion of OOD samples that are detected by the method. It is given by TPR $= \Pr\left(a(\mathbf{x}) > \gamma \,|\, Z = 1\right)$.

In order to evaluate the performance of our methods we will focus and report mainly the AUROC and the FPR $\downarrow$, we provide more detailed metrics and experiments in Sec. B.1.

**Area Under the Receiver Operating Characteristic curve (AUROC).** The Receiver Operating Characteristic curve is curve obtained by plotting the True positive rate against the False positive rate. The area under this curve is the probability that an in-distribution example $\mathbf{X}_{in}$ has an anomaly score higher than an OOD sample $\mathbf{x}_{out}$: AUROC$= \Pr(a(\mathbf{x}_{in}) > a(\mathbf{x}_{out}))$. It is given by $\gamma \mapsto \left(\Pr\left(a(\mathbf{x}) > \gamma \,|\, Z = 0\right), \Pr\left(a(\mathbf{x}) > \gamma \,|\, Z = 1\right)\right)$.

**False Positive Rate at $95\%$ True Positive Rate (FPR $\downarrow$).** We accept to allow only a given false positive rate $r$ corresponding to a defined level of safety and we want to know what share of positive samples we actually catch under this constraint. It leads to select a threshold $\gamma_r$ such that the corresponding TPR equals $r$. At this threshold, one then computes: $\Pr(a(\mathbf{x}) > \gamma_r \,|\, Z = 0)$ with $\gamma_r$ s.t. TPR$(\gamma_r) = r$. $r$ is chosen depending on the difficulty of the task at hand and the required level of safety.

For the sake of brevity, we present only AUROCand FPR $\downarrow$metrics in our aggregated results but we also used Detection error and Area Under the Precision-Recall curve metrics and those are presented in our full results section (Ap. F).

**Detection error**. It is simply the probability of miss-classification for a given True positive rate.

**Area Under the Precision-Recall curve (AUPR-IN/AUPR-OUT).** The Precision-Recall curve plots the recall (true detection rate) against the precision (actual proportion of OOD amongst the predicted OOD). The area under this curve $\gamma \mapsto \left(\Pr\left(Z = 1 \,|\, s(\mathbf{X}) \leqslant \gamma\right), \Pr\left(s(\mathbf{X}) \leqslant \gamma \,|\, Z = 1\right)\right)$ captures the trade-off between precision and recall made by the model. A high value represents a high precision and a high recall *i.e.* the detector captures most of the positive samples while having few False positives.

## B.2 Language pairs

| Model | IN data | OUT data |
|---|---|---|
| **Language shift** | | |
| DEU-ENG | Tatoeba DEU | News FR |
| DEU-ENG | Tatoeba DEU | Tatoeba NLD |
| SPA-ENG | Tatoeba SPA | News FR |
| SPA-ENG | Tatoeba SPA | Tatoeba CAT |
| SPA-ENG | Tatoeba SPA | Tatoeba POR |
| NLD-ENG | Tatoeba SPA | AFR |
| **Domain shift** | | |
| DEU-ENG | Tatoeba DEU | EMEA DEU |
| DEU-ENG | Tatoeba DEU | Eurparl DEU |
| DEU-ENG | Tatoeba DEU | EMEA DEU |
| DEU-ENG | Tatoeba DEU | Eurparl DEU |

Table 8: **Summary of models and studied shifts.**

## B.3 Dataset sizes

| Dataset Name | Size |
|---|---|
| Tatoeba AFR | 1373 |
| Tattoeba CAT | 1630 |
| Tatoeba DEU | 3000 |
| Tatoeba NLD | 3000 |
| Tatoeba POR | 3000 |
| Tatoebamt ES | 3000 |
| newscommentary DE | 3000 |
| newscommentary ES | 3000 |
| newscommentary FR | 6000 |
| newscommentary NL | 3000 |
| amazonreviewsmulti DE | 3000 |
| amazonreviewsmulti ES | 3000 |
| dailydialog default | 3000 |
| europarlbilingual DE | 3000 |
| europarlbilingual ES | 3000 |
| EMEA DE | 3000 |
| EMEA ES | 3000 |
| EMEA NL | 3000 |
| multiwoz | 3000 |
| silicone dydae | 3000 |
| silicone iemocap | 805 |
| silicone maptask | 2963 |
| silicone melds | 1109 |
| silicone mrda | 3000 |
| silicone oasis | 1513 |
| silicone sem | 485 |
| silicone swda | 3000 |

Table 9: Number of samples in each (test) datasets

## B.4 Samples

In Tab. 10 we provide examples of small shifts in translation between Spanish and Catalan and its impact on a spanish to english translation model.

## B.5 Dialog datasets

**Switchboard Dialog Act Corpus (SwDA)** is a corpus of telephonic conversations. The corpus provides labels, topic and speaker information (Stolcke et al., 2000).

**ICSI MRDA Corpus (MRDA)** contains transcript 75h of naturally occuring meetings involving more than 50 people (Shriberg et al., 2004).

**DaylyDialog Act Corpus (DyDA)** contains daily common communications between people, covering topic such as small talk, meteo or daily activities (Li et al., 2017).

**Interactive Emotional Dyadic Motion Capture IEMOCAP)**(Tripathi et al., 2018) consists of transcripts of improvisations or scripted scenarii supposed to outline the expression of emotions.

## B.6 Choices of models

To perform our experiments we needed models that were already well installed and deployed and that would also support OOD settings. For translation tasks, we needed specialized models for a notion of OOD to be easily defined. It would be indeed more hazardous to define a notion of OOD language when working with a multilingual model. The same is true for conversational models.

**Neural Machine Translation model.** We benchmark our OOD method on translation models provided by Helsinky NLP (Tiedemann and Thottingal, 2020) on several pairs of languages with large and small shifts. We extended the experiment to detect domain shifts. These models are indeed specialized in each language pair and are widely recognised in the neural machine translation field. For our experiments we used the testing set provided along these models, so we can consider that they have been fine-tuned over the same distribution.

**Conversational model.** We used a dialog-GPT (Zhang et al., 2019b) model fine-tuned on the Multi WOZ dataset as chat bot model. The fine-tuning on daily dialogue-type tasks ensures that the model is specialized, thus allowing us to get a good definition of samples not being in its range of expertise. Moreover, the choice of the architecture, DialogGPT, guarantees that our results are valid on a very common architecture.

| Source sentence | Expected translation | Translation | BLEU |
|---|---|---|---|
| A en Tom li agrada la tecnologia. | Tom likes technology. | Tom li likes technology. | 42.73 |
| Ací està la teua bossa. | Here is your bag. | Ací está la teua bossa. | 8.12 |
| Això et posarà en perill. | That'll put you in danger. | Això et posarà en perill. | 8.12 |
| A Londres hi han molts parcs bonics. | There are many beautiful parks in London. | To London hi han molts parcs bonics. | 6.57 |
| Aquest pa és molt deliciós. | This bread is very delicious. | Aquest pa és molt deliciós. | 8.12 |
| A tots els meus amics els agraden els videojocs. | All my friends like playing videogames. | A tots els meus amics els agrade els videojocs. | 4.20 |
| Açò és un peix. | This is a fish. | Aaaaaaaaaaaaaaaaaaa ... aaaaaaaaaaaaaaaaaaaaaaaaaa | 0.00 |
| Moltes felicitats! | Congratulations! | Moltes congrats! | 27.52 |
| Bon any nou! | Happy New Year! | Bon any nou! | 15.97 |
| Aquell que menteix, robarà. | He that will lie, will steal. | The one who's mindless, he'll steal. | 12.22 |
| Jo sóc qui té la clau. | I'm the one who has the key. | Jo soc qui te la clau. | 5.69 |
| En Tom surt a treballar cada matí a dos quarts de set. | Tom leaves for work at 6:30 every morning. | In Tom surt to pull each matí to two quarts of set. | 3.67 |
| Ell m'ha dit que la seva casa era embruixada. | He told me that his house was haunted. | Ell m'ha dit that the seva house was haunted. | 27.78 |
| Aquest és el lloc on va nèixer el meu pare. | This is the place where my father was born. | Aquest is the lloc on va nèixer el meu pare. | 8.30 |

Table 10: **Example of behaviours** of a language model trained to handle Spanish inputs on Catalan inputs.

## B.7 Generalization of existing OOD detectors to Sequence Generation

In this section, we extend classical OOD detection score to the conditional text generation settting. Common OOD detectors were built for classification tasks and we need to adapt them to conditional text generation. Our task can be viewed as a sequence of classification problems with a very large number of classes (the size of the vocabulary). We chose the most naive approach which consists of averaging the OOD scores over the sequence. We experimented with other aggregation such as the min/max or the standard deviation without getting interesting results.

**Likelihood Score** The most naive approach to build a OOD score is to rely solely on the log-likelihood of the sequence. For a conditioning $\mathbf{x}$ we define the log-likelyhood score by $a_L(\mathbf{x}) = -\sum_{t=0}^{|\hat{\mathbf{y}}|-1} \log \mathbf{p}_\theta(\hat{\mathbf{y}}_{t+1}|\mathbf{x}, \hat{\mathbf{y}}_{\leqslant t})$. The likelihood is the same as the perplexity.

**Average Maximum Softmax Probability score** The maximum softmax probability (Hendrycks and Gimpel, 2017) takes the probability of the mode of the categorical distribution as score of OOD. We extend thise definition in the case of sequence of probability distribution by averaging this score along the sequence. For a given conditioning $\mathbf{x}$, we define the average MSP score $a_{\text{MSP}}(\mathbf{x}) = \frac{1}{|\hat{\mathbf{y}}|} \sum_{t=1}^{|\hat{\mathbf{y}}|} \max_{i \in [|0,K|]} \mathbf{p}_\theta^T(i|\mathbf{x}, \hat{\mathbf{y}}_{\leqslant t}))$. While it is closely linked to uncertainty measures it discards most of the information contained in the probability distribution. It discards the whole probability distribution. We claim that much more information can be retrieve by studying the whole distribution.

**Average Energy score** We extend the definition of the energy score described in (Liu et al., 2020) to a sequence of probability distributions by averaging the score along the sequence. For a given conditioning $\mathbf{x}$ and a temperature $T$ we define the average energy of the sequence: $a_E(\mathbf{x}) \triangleq -\frac{T}{|\hat{\mathbf{y}}|} \sum_{t=1}^{|\hat{\mathbf{y}}|} \log \sum_i^{|\Omega|} e^{f_\theta(\mathbf{x}, \hat{\mathbf{y}}_{\leqslant t})_i/T}$. It corresponds to the normalization term of the softmax function applied on the logits. While it takes into account the whole distribution, it only takes into account the amount of unnormalized mass before normalization without attention to how this mass is distributed along the features.

**Mahalanobis distance** Following (Lee et al., 2018a; Colombo et al., 2022a) compute the Mahalanobis matrice based on the samples of a given reference set $\mathcal{R}$. In our case we are using encoder-decoder models we use the output of the last hidden layer of the encoder as embedding. Let's denote $\phi(\mathbf{x})$ this embedding for a conditioning $\mathbf{x}$. Let $\mu$ and $\Sigma$ be respectively, the mean and the covariance of these embedding on the reference set. We define $a_{\text{M}}(\mathbf{x}) = \left(1 + (\phi(\mathbf{x}) - \mu)^\top \Sigma^{-1} (\phi(\mathbf{x}) - \mu)\right)^{-1}$.

## B.8 Computational budget

We had a budget of 20000h on NVIDIA V100 GPU. While this is an important number it was used to compute the benchmarks over many pairs and languages. In practice our OOD detectors do not require much addition computation overhead since they only rely on the probability distributions already output by the models.

## B.9 Towards an interpretable decision

An important dimension of fostering adoption is the ability to verify the decision taken by the automatic system. RAINPROOF offers a step in this direction when used with references: for each input sample, RAINPROOF finds the closest sample (in the sense of the Information Projection) in the reference set to take its decision. We present in Tab. 11 some OOD samples along with their translation scores, projection scores, and their projection on the refer-

Table 11: OOD inputs, their translations and projections onto the reference set. The first 2 are far from the reference set and not well translated whereas the next 2 are very close to the reference set and well translated. We can, for that matter, notice that the projection is quite close to the input sentence grammatically speaking.

| | |
|---|---|
| Source | Ahir a la nit vàrem treballar fins a les deu. |
| Ground truth | Last night we worked until 10 p.m. |
| Generated | Ahir a la nit vàrem treballar fins a les deu.    BLEU 3.75 |
| $\mathbf{p}^\star(\mathbf{x})$ | Dar gato por liebre.    **Score** 1.23 |
| Source | Aquesta cola s'ha esbravat i no té bon gust. |
| Ground-truth | This cola has lost its fizz and doesn't taste any good. |
| Generated | This tail s'ha esbravat i no tea bon gust.    BLEU 4.09 |
| $\mathbf{p}^\star(\mathbf{x})$ | Esta cuchara es de té.    **Score** 1.14 |
| source | Aquesta és una carta molt estranya. |
| Ground-truth | This is a very strange letter. |
| Generated | This is a molt estranya card.    BLEU 26.27 |
| $\mathbf{p}^\star(\mathbf{x})$ | Este carro es chiquito.    **Score** 0.74 |
| source | Austràlia no és Àustria. |
| Ground-truth | Australia isn't Austria. |
| Generated | Austràlia is not Austria.    BLEU 21.86 |
| $\mathbf{p}^\star(\mathbf{x})$ | La vida no es fácil.    **Score** 0.82 |

ence set. We notice that, in general, sentences that are close to the reference set, and whose projection has a close meaning, are better handled by $f_\theta$. Therefore, one can visually interpret the prediction of RAINPROOF, and validate it. This observation further validates our method.

# C    Scaling to larger models

In order to validate our results we perform experiments on larger and general-purpose models such as BloomZ (Muennighoff et al., 2022), NLLB (Team et al., 2022) and the Facebook WMT16 submission (Ng et al., 2020).

| | | | AUROC | FPR ↓ | AUPR-IN | AUPR-OUT | Err |
|---|---|---|---|---|---|---|---|
| | Ours | $a_{D_\alpha}$ | 0.72 | 0.80 | 0.54 | 0.79 | 0.51 |
| $\$_0$ | Bas. | $a_{CQE}$ | 0.65 | 0.95 | 0.74 | 0.59 | 0.52 |
| | | $a_L$ | 0.47 | 0.89 | 0.59 | 0.50 | 0.43 |
| | Ours. | $a_{D^*}$ | 0.74 | 0.87 | 0.80 | 0.68 | 0.34 |
| | | $a_{FR*}$ | 0.74 | 0.88 | 0.80 | 0.67 | 0.35 |
| $\$_1$ | | $a_C$ | 0.70 | 0.90 | 0.80 | 0.52 | 0.38 |
| | Bas. | $a_D$ | 0.64 | 0.89 | 0.54 | 0.67 | 0.53 |
| | | $a_M$ | 0.62 | 0.81 | 0.66 | 0.55 | 0.38 |

Table 12: Performance in OOD detection on large translation model Facebook WMT-19 submission. (RUS-DEU).

| | | | AUROC | FPR ↓ | AUPR-IN | AUPR-OUT | Err |
|---|---|---|---|---|---|---|---|
| | Ours | $a_{FR}$ | 0.57 | 0.93 | 0.50 | 0.64 | 0.50 |
| | | $a_{D_\alpha}$ | 0.58 | 0.92 | 0.52 | 0.63 | 0.51 |
| $\$_0$ | | $a_{MSP}$ | **0.59** | **0.90** | **0.64** | **0.52** | **0.42** |
| | Bas. | $a_E$ | 0.51 | 0.97 | 0.47 | 0.55 | 0.57 |
| | | $a_L$ | 0.54 | 0.90 | 0.44 | 0.62 | 0.53 |
| | | $a_{D_\alpha^*}$ | **0.71** | **0.82** | **0.73** | **0.64** | **0.39** |
| $\$_1$ | $\$_1$ | $a_{FR*}$ | 0.70 | 0.82 | 0.73 | 0.64 | 0.40 |
| | Bas. | $a_M$ | 0.66 | 0.89 | 0.74 | 0.56 | 0.42 |

Table 13: OOD detection on BloomZ using German as an OOD language for the instruction model.

## C.1    Negative results on NLLB

By the very definition of the No-Language-Left-Behind model, it should be particularly hard to find OOD language to benchmark on. The model still requires special token to be set in the sequence to define the source and target languages. We tried to apply our OOD detection methods to situations where the presented language does not correspond to the source language set by the special token. We found that in this scenario the likelihood was by far the best discriminator of OOD samples. It can be explained by the fact that our inputs are not actually OOD, they are just not consistent with the source language token, but the model is still well calibrated overall on these inputs.

| | | | AUROC | FPR ↓ | AUPR-IN | AUPR-OUT | Err |
|---|---|---|---|---|---|---|---|
| | Ours | $a_{FR}$ | 0.50 | 1.00 | 0.75 | 0.75 | 0.51 |
| | | $a_{D_\alpha}$ | 0.57 | 0.95 | 0.60 | 0.62 | 0.50 |
| $\$_0$ | | $a_{MSP}$ | 0.71 | 0.80 | 0.71 | 0.68 | 0.42 |
| | Bas. | $a_E$ | 0.53 | 0.97 | 0.54 | 0.50 | 0.51 |
| | | $a_L$ | 0.80 | 0.59 | 0.78 | 0.79 | 0.32 |
| $\$_1$ | Ours | $a_{FR*}$ | 0.54 | 0.93 | 0.57 | 0.49 | 0.46 |
| | Bas. | $a_M$ | 0.62 | 0.89 | 0.67 | 0.54 | 0.42 |

Table 14: Performance in OOD detection for NLLB.

# D    Additional OOD features-based baselines

To further support the point that features-based detectors have important flaws when it comes to text generation we compare our best performing OOD score to SOTA OOD detectors in text such as the DataDepth ($a_D$) (Colombo et al., 2022a) and the Maximum Cosine Projection ($a_C$) (Zhou et al., 2021).

|  |  |  | AUROC | FPR ↓ | AUPR-IN | AUPR-OUT | Err |
|---|---|---|---|---|---|---|---|
| $\mathbb{s}_0$ | Ours | $a_{D_\alpha}$ | 0.85 | 0.66 | 0.69 | 0.91 | 0.47 |
|  | Bas. | $a_{\text{CQE}}$ | 0.76 | 0.51 | 0.78 | 0.71 | 0.20 |
|  |  | $a_L$ | 0.81 | 0.55 | 0.86 | 0.69 | 0.20 |
| $\mathbb{s}_1$ | Bas. | $a_C$ | 0.61 | 0.95 | 0.82 | 0.36 | 0.32 |
|  |  | $a_D$ | 0.57 | 0.93 | 0.35 | 0.75 | 0.67 |
|  |  | $a_M$ | 0.70 | 0.80 | 0.46 | 0.84 | 0.58 |

Table 15: Comparison of our best detector $a_{D_\alpha}$ against SOTA features based-ood detectors on close language shifts.

# E    Parameters tuning

Detectors depend on their anomaly score to make decisions, and these scores can be parametric. First of all, soft probability-based scores depend on the soft probability distribution and its scaling. Therefore, the temperature is a crucial parameter to tune to get the most performance. While a small temperature makes the distribution pickier, a higher value spreads the probability mass along the classes. Moreover, the Rényi divergence depends on a factor $\alpha$. We provide here further results and analysis of those parameters on our results.

In Fig. 3, we analyse the impact of the temperature and $\alpha$ parameter for our Renyi-Negentropy score. Consistently with results for the information projection we find that the tail of the distribution is important to ensure good detection of OOD samples for all language shifts. A temperature higher than 2 and lower values of $\alpha$ yield the best results. We recommend using $\alpha = 0.5$ with a temperature of 2.

We found that our $a_{D_\alpha}$ score, the Rényi negentropy is more stable concerning the temperature and the considered datasets and shifts than the energy-based OOD score and the MSP score. Indeed, in Fig. 4, we show that the baselines do not behave consistently across datasets when the temperature changes. This is a problem when deploying these scores in production. Indeed, we cannot fit a temperature for each possible type of shift or OOD samples. By contrast, there exist sets of parameters (temperature and $\alpha$) for which our negentropy-based scores perform consistently across different shifts.

# F    Performance of our detectors in OOD detection

## F.1    Importance of tails' distributions

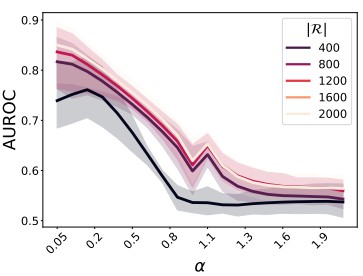

Figure 5: **Impact of $\alpha$ on the performance** of the Rényi information projection for dialog shifts detection. A smaller $\alpha$ increases the weight of the tail of the distribution. An $\alpha$ of 0 would consist in counting the number of the common non zero elements.

Our results show that, when it comes to domain shift (domain shifts in translation or dialog shifts), reference-based detectors are required to obtain good results. They also show that, the more these detectors take into account the tail of the distributions, the better they are, as displayed in Fig. 5. We find that low values of $\alpha$ (near 0) yields better results with the Rényi Information projection $a_{D_\alpha^*}$. It suggests that the tail of the distributions used during text generation carries context information and insights on the processed texts. Such results are consistent with findings of recent works in the context of automatic evaluation of text generation (Colombo et al., 2022b).

## F.2    Summary of our results

In Fig. 6 we present the different performance levels of all the detectors we studied. We can see that in every task our detectors outperform the baselines but also that in dialog shift, while the Mahalanobis distance outperform clearly our detectors for $\mathbb{s}_0$, they still outperform baselines for their scenario by far.

## F.3    Detailed results of OOD detection performances

In this section, we present the performances of our OOD detectors on each detailed tasks, *i.e.* for each pair of IN and OOD data with all the considered metrics. Our metrics outperform other OOD detectors baselines in almost all scenarios.

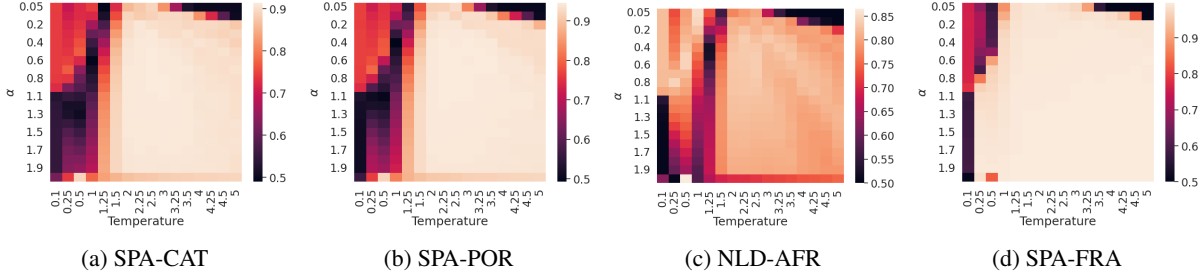

(a) SPA-CAT      (b) SPA-POR      (c) NLD-AFR      (d) SPA-FRA

Figure 3: Effect of the temperature and $\alpha$ parameter for $a_{D_\alpha}$ on the performance on OOD detection in terms of AUROC.

Table 16: Detailed results of the performances of our OOD detectors on different language shifts. The first language of the pair is the reference language of the model and the second one is the studied shift.

| Scenario | | Score | AUPR-IN | AUPR-OUT | AUROC | ERR | f1 | FPR | precision | recall |
|---|---|---|---|---|---|---|---|---|---|---|
| $s_0$ | Ours | $a_{D_\alpha}$ | **0.99** | **1.00** | **1.00** | 0.02 | **0.83** | **0.01** | **0.71** | **1.00** |
| | | $a_{FR}$ | 0.98 | **1.00** | 0.99 | 0.03 | **0.83** | 0.02 | **0.71** | 1.00 |
| | Baselines | $a_E$ | 0.96 | 0.96 | 0.97 | 0.05 | 0.82 | 0.05 | **0.71** | 1.00 |
| | | $a_L$ | 0.61 | 0.84 | 0.78 | 0.53 | 0.63 | 0.76 | 0.60 | 0.65 |
| | | $a_{MSP}$ | 0.96 | 0.99 | 0.98 | 0.05 | 0.82 | 0.05 | **0.71** | 1.00 |
| | | $a_{CQE}$ | 0.50 | 0.76 | 0.62 | 0.60 | 0.34 | 0.87 | 0.42 | 0.29 |
| $s_1$ | Ours | $a_{D_\alpha^*}$ | **1.00** | **1.00** | **1.00** | 0.02 | **0.83** | **0.00** | **0.71** | **1.00** |
| | | $a_{D_{KL}^*}$ | 0.96 | 0.99 | 0.99 | 0.03 | **0.83** | 0.03 | **0.71** | 0.99 |
| | | $a_{D_\alpha^{mean}}$ | **1.00** | **1.00** | **1.00** | 0.02 | **0.83** | **0.00** | **0.71** | 0.99 |
| | | $a_{D_{KL}^{mean}}$ | 0.99 | 0.98 | 0.99 | 0.02 | **0.83** | 0.00 | **0.71** | 1.00 |
| | | $a_{FR^*}$ | **1.00** | **1.00** | **1.00** | 0.02 | **0.83** | **0.00** | **0.71** | **1.00** |
| | | $a_{FR^{mean}}$ | 0.99 | 0.97 | 0.98 | 0.02 | 0.82 | 0.01 | **0.71** | 0.98 |
| | Baselines | $a_C$ | 0.39 | 0.74 | 0.60 | 0.64 | 0.42 | 0.94 | 0.42 | 0.42 |
| | | $a_M$ | 0.64 | 0.91 | 0.82 | 0.33 | 0.63 | 0.47 | 0.62 | 0.64 |

(a) deu-nld

| Scenario | | Score | AUPR-IN | AUPR-OUT | AUROC | ERR | f1 | FPR | precision | recall |
|---|---|---|---|---|---|---|---|---|---|---|
| $s_0$ | Ours | $a_{D_\alpha}$ | **0.80** | **0.97** | **0.91** | 0.33 | **0.67** | 0.41 | **0.54** | **1.00** |
| | | $a_{FR}$ | **0.80** | 0.96 | **0.91** | 0.37 | **0.67** | 0.46 | **0.54** | 0.87 |
| | Baselines | $a_E$ | 0.61 | 0.89 | 0.77 | 0.71 | 0.54 | 0.88 | 0.46 | **1.00** |
| | | $a_L$ | 0.55 | 0.89 | 0.75 | 0.70 | 0.35 | 0.88 | 0.21 | **1.00** |
| | | $a_{MSP}$ | 0.61 | 0.90 | 0.73 | 0.70 | 0.41 | 0.87 | 0.37 | **1.00** |
| | | $a_{CQE}$ | 0.27 | 0.76 | 0.51 | 0.80 | 0.25 | 1.00 | 0.23 | 0.27 |
| $s_1$ | Ours | $a_{D_\alpha^*}$ | 0.48 | 0.87 | 0.70 | 0.73 | 0.50 | 0.91 | 0.44 | 0.57 |
| | | $a_{D_{KL}^*}$ | 0.39 | 0.85 | 0.66 | 0.74 | 0.43 | 0.93 | 0.39 | 0.48 |
| | | $a_{D_\alpha^{mean}}$ | **0.58** | **0.89** | **0.77** | 0.74 | **0.54** | 0.92 | **0.47** | 0.65 |
| | | $a_{D_{KL}^{mean}}$ | 0.45 | 0.79 | 0.63 | 0.79 | 0.45 | 1.00 | 0.41 | 0.50 |
| | | $a_{FR^*}$ | 0.45 | 0.86 | 0.69 | 0.74 | 0.46 | 0.92 | 0.41 | 0.52 |
| | | $a_{FR^{mean}}$ | 0.41 | 0.76 | 0.56 | 0.79 | 0.00 | 0.99 | 0.00 | 0.00 |
| | Baselines | $a_C$ | 0.33 | 0.84 | 0.63 | 0.75 | 0.40 | 0.94 | 0.33 | 0.50 |
| | | $a_M$ | 0.37 | **0.89** | 0.71 | **0.65** | 0.46 | **0.81** | 0.41 | 0.51 |

(b) spa-cat

| Scenario | | Score | AUPR-IN | AUPR-OUT | AUROC | ERR | f1 | FPR | precision | recall |
|---|---|---|---|---|---|---|---|---|---|---|
| $s_0$ | Ours | $a_{D_\alpha}$ | **0.67** | **0.95** | **0.87** | 0.55 | 0.55 | **0.67** | 0.45 | **1.00** |
| | | $a_{FR}$ | 0.64 | 0.94 | 0.83 | 0.58 | 0.55 | 0.70 | 0.45 | 0.72 |
| | Baselines | $a_E$ | 0.65 | 0.94 | 0.84 | 0.62 | **0.57** | 0.74 | **0.46** | 1.00 |
| | | $a_L$ | 0.64 | 0.94 | 0.84 | 0.65 | 0.31 | 0.79 | 0.19 | 1.00 |
| | | $a_{MSP}$ | 0.61 | 0.94 | 0.83 | 0.62 | 0.31 | 0.75 | 0.19 | 1.00 |
| | | $a_{CQE}$ | 0.29 | 0.85 | 0.58 | 0.76 | 0.27 | 0.92 | 0.28 | 0.26 |
| $s_1$ | Ours | $a_{D_\alpha^*}$ | 0.48 | **0.94** | 0.79 | **0.47** | 0.47 | **0.57** | 0.40 | 0.57 |
| | | $a_{D_{KL}^*}$ | 0.37 | **0.94** | 0.75 | 0.49 | 0.40 | 0.59 | 0.35 | 0.47 |
| | | $a_{D_\alpha^{mean}}$ | **0.66** | 0.93 | **0.83** | 0.69 | 0.39 | 0.84 | 0.34 | 0.46 |
| | | $a_{D_{KL}^{mean}}$ | 0.29 | 0.82 | 0.55 | 0.80 | 0.00 | 0.97 | 0.00 | 0.00 |
| | | $a_{FR^*}$ | 0.47 | **0.94** | 0.78 | **0.47** | 0.46 | **0.57** | 0.39 | 0.56 |
| | | $a_{FR^{mean}}$ | 0.52 | 0.90 | 0.75 | 0.74 | 0.00 | 0.90 | 0.00 | 0.00 |
| | Baselines | $a_C$ | 0.23 | 0.86 | 0.58 | 0.77 | 0.29 | 0.94 | 0.22 | 0.40 |
| | | $a_M$ | 0.35 | 0.92 | 0.74 | 0.61 | 0.45 | 0.74 | 0.39 | 0.55 |

(c) nld-afr

| Scenario | | Score | AUPR-IN | AUPR-OUT | AUROC | ERR | f1 | FPR | precision | recall |
|---|---|---|---|---|---|---|---|---|---|---|
| $s_0$ | Ours | $a_{D_\alpha}$ | **0.99** | **1.00** | **1.00** | 0.02 | **0.83** | **0.01** | **0.71** | **1.00** |
| | | $a_{FR}$ | 0.97 | 0.99 | 0.99 | 0.04 | **0.83** | 0.04 | **0.71** | **1.00** |
| | Baselines | $a_E$ | 0.97 | 0.97 | 0.98 | 0.04 | 0.82 | 0.03 | **0.71** | 1.00 |
| | | $a_L$ | 0.68 | 0.83 | 0.77 | 0.58 | 0.61 | 0.85 | 0.60 | 0.63 |
| | | $a_{MSP}$ | 0.98 | 0.99 | 0.99 | 0.04 | **0.83** | 0.04 | **0.71** | 1.00 |
| | | $a_{CQE}$ | 0.48 | 0.85 | 0.68 | 0.38 | 0.37 | 0.55 | 0.42 | 0.33 |
| $s_1$ | Ours | $a_{D_\alpha^*}$ | **1.00** | **1.00** | **1.00** | 0.02 | **0.83** | **0.00** | **0.71** | **1.00** |
| | | $a_{D_{KL}^*}$ | **1.00** | **1.00** | 0.99 | 0.02 | **0.83** | 0.01 | **0.71** | **1.00** |
| | | $a_{D_\alpha^{mean}}$ | **1.00** | **1.00** | **1.00** | 0.01 | **0.83** | **0.00** | **0.71** | **1.00** |
| | | $a_{D_{KL}^{mean}}$ | **1.00** | **1.00** | **1.00** | 0.02 | **0.83** | **0.00** | **0.71** | 0.99 |
| | | $a_{FR^*}$ | **1.00** | **1.00** | **1.00** | 0.02 | **0.83** | **0.00** | **0.71** | **1.00** |
| | | $a_{FR^{mean}}$ | 0.99 | 0.99 | 0.99 | 0.02 | **0.83** | **0.00** | **0.71** | 0.99 |
| | Baselines | $a_C$ | 0.38 | 0.73 | 0.61 | 0.66 | 0.45 | 0.97 | 0.43 | 0.46 |
| | | $a_M$ | 0.68 | 0.92 | 0.84 | 0.29 | 0.66 | 0.41 | 0.63 | 0.69 |

(d) deu-fra

| Scenario | | Score | AUPR-IN | AUPR-OUT | AUROC | ERR | f1 | FPR | precision | recall |
|---|---|---|---|---|---|---|---|---|---|---|
| $s_0$ | Ours | $a_{D_\alpha}$ | **1.00** | **1.00** | **1.00** | 0.02 | **0.83** | **0.00** | **0.71** | **1.00** |
| | | $a_{FR}$ | 0.99 | **1.00** | **1.00** | 0.02 | **0.83** | 0.01 | **0.71** | 1.00 |
| | Baselines | $a_E$ | 0.98 | 0.98 | 0.98 | 0.04 | **0.83** | 0.03 | **0.71** | 1.00 |
| | | $a_L$ | 0.53 | 0.83 | 0.73 | 0.53 | 0.53 | 0.77 | 0.54 | 0.52 |
| | | $a_{MSP}$ | 0.97 | 0.99 | 0.99 | 0.04 | 0.82 | 0.04 | **0.71** | 1.00 |
| | | $a_{CQE}$ | 0.58 | 0.87 | 0.74 | 0.38 | 0.50 | 0.55 | 0.50 | 0.50 |
| $s_1$ | Ours | $a_{D_\alpha^*}$ | 0.99 | **1.00** | **1.00** | 0.02 | **0.83** | 0.01 | **0.71** | **1.00** |
| | | $a_{D_{KL}^*}$ | 0.90 | 0.99 | 0.97 | 0.07 | **0.83** | 0.07 | **0.71** | 0.99 |
| | | $a_{D_\alpha^{mean}}$ | **1.00** | **1.00** | **1.00** | 0.02 | **0.83** | **0.00** | **0.71** | **1.00** |
| | | $a_{D_{KL}^{mean}}$ | 0.99 | 0.99 | 0.99 | 0.02 | **0.83** | 0.01 | **0.71** | 0.99 |
| | | $a_{FR^*}$ | 0.99 | **1.00** | **1.00** | 0.02 | **0.83** | 0.01 | **0.71** | **1.00** |
| | | $a_{FR^{mean}}$ | 0.99 | 0.98 | 0.99 | 0.03 | 0.82 | 0.01 | **0.71** | 0.98 |
| | Baselines | $a_C$ | 0.41 | 0.71 | 0.57 | 0.65 | 0.00 | 0.96 | 0.00 | 0.00 |
| | | $a_M$ | 0.81 | 0.97 | 0.93 | 0.13 | 0.82 | 0.17 | **0.71** | 0.97 |

(e) spa-fra

| Scenario | | Score | AUPR-IN | AUPR-OUT | AUROC | ERR | f1 | FPR | precision | recall |
|---|---|---|---|---|---|---|---|---|---|---|
| $s_0$ | Ours | $a_{D_\alpha}$ | 0.91 | **0.97** | **0.95** | 0.20 | **0.80** | 0.27 | **0.70** | **1.00** |
| | | $a_{FR}$ | **0.92** | **0.97** | **0.95** | 0.19 | **0.80** | 0.26 | **0.70** | 0.93 |
| | Baselines | $a_E$ | 0.71 | 0.84 | 0.79 | 0.58 | 0.64 | 0.85 | 0.62 | 1.00 |
| | | $a_L$ | 0.69 | 0.83 | 0.77 | 0.58 | 0.50 | 0.85 | 0.33 | **1.00** |
| | | $a_{MSP}$ | 0.67 | 0.86 | 0.74 | 0.58 | 0.55 | 0.84 | 0.55 | **1.00** |
| | | $a_{CQE}$ | 0.46 | 0.67 | 0.54 | 0.67 | 0.30 | 0.98 | 0.38 | 0.24 |
| $s_1$ | Ours | $a_{D_\alpha^*}$ | 0.74 | 0.83 | 0.78 | 0.60 | 0.65 | 0.87 | 0.63 | 0.68 |
| | | $a_{D_{KL}^*}$ | 0.67 | 0.82 | 0.76 | 0.62 | 0.63 | 0.90 | 0.61 | 0.64 |
| | | $a_{D_\alpha^{mean}}$ | **0.80** | 0.87 | **0.84** | 0.57 | **0.70** | 0.82 | **0.65** | 0.75 |
| | | $a_{D_{KL}^{mean}}$ | 0.68 | 0.72 | 0.70 | 0.68 | 0.61 | 0.99 | 0.61 | 0.62 |
| | | $a_{FR^*}$ | 0.72 | 0.83 | 0.78 | 0.60 | 0.63 | 0.88 | 0.61 | 0.66 |
| | | $a_{FR^{mean}}$ | 0.48 | 0.60 | 0.48 | 0.68 | 0.00 | 1.00 | 0.00 | 0.00 |
| | Baselines | $a_C$ | 0.48 | 0.76 | 0.66 | 0.64 | 0.52 | 0.94 | 0.49 | 0.55 |
| | | $a_M$ | 0.61 | **0.91** | 0.83 | 0.38 | 0.69 | **0.55** | **0.65** | 0.74 |

(f) spa-por

Table 17: Detailed results of the performances of our OOD detectors on different domain shifts. For Spanish (spa) and German (de), we present two domains shifts: Technical medical (EMEA) data and legal parlementary texts (parl) against common language embodied by the Tatoeba dataset (tat).

| Scenario | | Score | AUPR-IN | AUPR-OUT | AUROC | ERR | f1 | FPR | precision | recall |
|---|---|---|---|---|---|---|---|---|---|---|
| $s_0$ | Ours | $a_{D_\alpha}$ | **0.90** | **0.76** | **0.86** | **0.43** | 0.81 | **0.82** | 0.80 | 1.00 |
| | | $a_{FR}$ | 0.87 | 0.73 | 0.81 | 0.46 | 0.77 | 0.86 | 0.79 | 0.75 |
| | Baselines | $a_E$ | 0.88 | 0.75 | 0.83 | 0.49 | 0.78 | 0.93 | 0.79 | 1.00 |
| | | $a_L$ | 0.86 | 0.73 | 0.82 | 0.48 | 0.76 | 0.91 | 0.77 | 0.74 |
| | | $a_{MSP}$ | 0.89 | **0.76** | 0.85 | 0.44 | 0.79 | 0.84 | 0.80 | 1.00 |
| $s_1$ | Ours | $a_{D_\alpha^*}$ | **0.90** | 0.88 | **0.90** | 0.25 | 0.82 | 0.45 | 0.81 | 0.86 |
| | | $a_{D_{KL}^*}$ | 0.89 | 0.83 | 0.88 | 0.33 | 0.82 | 0.62 | 0.80 | 0.83 |
| | | $a_{D_\alpha^{mean}}$ | 0.88 | 0.73 | 0.84 | 0.47 | 0.78 | 0.89 | 0.79 | 0.77 |
| | | $a_{D_{KL}^{mean}}$ | 0.87 | 0.73 | 0.83 | 0.48 | 0.77 | 0.91 | 0.79 | 0.75 |
| | | $a_{FR^*}$ | 0.85 | 0.70 | 0.80 | 0.50 | 0.74 | 0.94 | 0.77 | 0.72 |
| | | $a_{FR^{mean}}$ | 0.86 | 0.67 | 0.80 | 0.52 | 0.75 | 0.98 | 0.78 | 0.72 |
| | Baselines | $a_C$ | 0.88 | 0.89 | **0.90** | 0.25 | 0.00 | 0.46 | 0.00 | 0.00 |
| | | $a_M$ | 0.87 | **0.90** | 0.89 | **0.22** | 0.81 | **0.39** | 0.80 | 0.81 |

(a) deu:news-EMEA

| Scenario | | Score | AUPR-IN | AUPR-OUT | AUROC | ERR | f1 | FPR | precision | recall |
|---|---|---|---|---|---|---|---|---|---|---|
| $s_0$ | Ours | $a_{D_\alpha}$ | **0.75** | **0.75** | **0.76** | **0.41** | 0.67 | **0.76** | 0.67 | 1.00 |
| | | $a_{FR}$ | 0.66 | 0.67 | 0.70 | 0.44 | 0.45 | 0.84 | 0.64 | 0.35 |
| | Baselines | $a_E$ | **0.75** | **0.75** | 0.71 | 0.44 | 0.67 | 0.83 | 0.69 | 1.00 |
| | | $a_L$ | 0.68 | 0.61 | 0.68 | 0.49 | 0.67 | 0.94 | 0.50 | 1.00 |
| | | $a_{MSP}$ | **0.75** | **0.75** | 0.71 | 0.45 | 0.67 | 0.86 | 0.50 | 1.00 |
| $s_1$ | Ours | $a_{D_\alpha^*}$ | 0.66 | 0.65 | 0.68 | 0.44 | 0.00 | 0.84 | 0.83 | 0.00 |
| | | $a_{D_{KL}^*}$ | **0.67** | 0.63 | 0.67 | 0.48 | 0.00 | 0.90 | 0.00 | 0.00 |
| | | $a_{D_\alpha^{mean}}$ | **0.67** | **0.67** | **0.70** | 0.44 | 0.00 | 0.82 | 0.00 | 0.00 |
| | | $a_{D_{KL}^{mean}}$ | 0.66 | 0.65 | 0.69 | 0.45 | 0.00 | 0.85 | 0.00 | 0.00 |
| | | $a_{FR^*}$ | 0.62 | 0.64 | 0.65 | 0.45 | 0.00 | 0.85 | 0.00 | 0.00 |
| | | $a_{FR^{mean}}$ | 0.65 | **0.67** | 0.69 | **0.43** | 0.00 | **0.80** | 0.00 | 0.00 |
| | Baselines | $a_C$ | 0.57 | 0.61 | 0.60 | 0.46 | 0.27 | 0.87 | 0.47 | 0.19 |
| | | $a_M$ | 0.62 | 0.66 | 0.66 | 0.44 | 0.00 | 0.83 | 0.00 | 0.00 |

(b) spa:news-parl

| Scenario | | Score | AUPR-IN | AUPR-OUT | AUROC | ERR | f1 | FPR | precision | recall |
|---|---|---|---|---|---|---|---|---|---|---|
| $s_0$ | Ours | $a_{D_\alpha}$ | **0.75** | **0.75** | **0.71** | **0.41** | 0.67 | **0.78** | 0.66 | 1.00 |
| | | $a_{FR}$ | 0.61 | 0.65 | 0.65 | 0.45 | 0.42 | 0.84 | 0.61 | 0.32 |
| | Baselines | $a_E$ | **0.75** | **0.75** | 0.68 | 0.45 | 0.67 | 0.85 | 0.66 | 1.00 |
| | | $a_L$ | 0.63 | 0.58 | 0.64 | 0.51 | 0.67 | 0.96 | 0.50 | 1.00 |
| | | $a_{MSP}$ | **0.75** | **0.75** | 0.68 | 0.46 | 0.67 | 0.86 | 0.51 | 1.00 |
| $s_1$ | Ours | $a_{D_\alpha^*}$ | **0.69** | **0.66** | **0.68** | 0.43 | 0.30 | 0.81 | 0.80 | 0.22 |
| | | $a_{D_{KL}^*}$ | 0.66 | 0.64 | **0.68** | 0.46 | 0.00 | 0.88 | 0.00 | 0.00 |
| | | $a_{D_\alpha^{mean}}$ | 0.65 | 0.65 | **0.68** | 0.45 | 0.00 | 0.86 | 0.00 | 0.00 |
| | | $a_{D_{KL}^{mean}}$ | 0.65 | 0.65 | **0.68** | 0.45 | 0.00 | 0.85 | 0.00 | 0.00 |
| | | $a_{FR^*}$ | 0.63 | 0.64 | 0.66 | 0.45 | 0.00 | 0.86 | 0.00 | 0.00 |
| | | $a_{FR^{mean}}$ | 0.64 | 0.64 | 0.67 | 0.45 | 0.00 | 0.86 | 0.00 | 0.00 |
| | Baselines | $a_C$ | 0.52 | **0.66** | 0.59 | 0.41 | 0.40 | **0.78** | 0.52 | 0.33 |
| | | $a_M$ | 0.58 | 0.61 | 0.62 | 0.47 | 0.00 | 0.89 | 0.00 | 0.00 |

(c) deu:news-parl

| Scenario | | Score | AUPR-IN | AUPR-OUT | AUROC | ERR | f1 | FPR | precision | recall |
|---|---|---|---|---|---|---|---|---|---|---|
| $s_0$ | Ours | $a_{D_\alpha}$ | **0.92** | **0.81** | **0.89** | **0.37** | 0.83 | **0.70** | 0.81 | 1.00 |
| | | $a_{FR}$ | 0.89 | 0.75 | 0.85 | 0.44 | 0.79 | 0.82 | 0.80 | 0.78 |
| | Baselines | $a_E$ | 0.90 | 0.77 | 0.86 | 0.44 | 0.80 | 0.83 | 0.80 | 1.00 |
| | | $a_L$ | 0.86 | 0.73 | 0.82 | 0.47 | 0.76 | 0.89 | 0.77 | 0.74 |
| | | $a_{MSP}$ | 0.90 | 0.80 | 0.87 | 0.41 | 0.81 | 0.77 | 0.80 | 1.00 |
| $s_1$ | Ours | $a_{D_\alpha^*}$ | 0.88 | **0.85** | **0.88** | 0.29 | 0.81 | **0.54** | 0.80 | 0.82 |
| | | $a_{D_{KL}^*}$ | 0.89 | 0.83 | **0.88** | 0.32 | 0.81 | 0.59 | 0.80 | 0.82 |
| | | $a_{D_\alpha^{mean}}$ | **0.90** | 0.77 | 0.86 | 0.44 | 0.79 | 0.83 | 0.80 | 0.79 |
| | | $a_{D_{KL}^{mean}}$ | 0.88 | 0.75 | 0.84 | 0.45 | 0.77 | 0.85 | 0.79 | 0.75 |
| | | $a_{FR^*}$ | 0.81 | 0.66 | 0.76 | 0.50 | 0.70 | 0.94 | 0.76 | 0.65 |
| | | $a_{FR^{mean}}$ | 0.87 | 0.70 | 0.81 | 0.49 | 0.75 | 0.94 | 0.78 | 0.72 |
| | Baselines | $a_C$ | 0.67 | 0.59 | 0.64 | 0.49 | 0.00 | 0.94 | 0.00 | 0.00 |
| | | $a_M$ | 0.81 | 0.83 | 0.83 | 0.31 | 0.75 | 0.58 | 0.78 | 0.72 |

(d) spa:news-EMEA

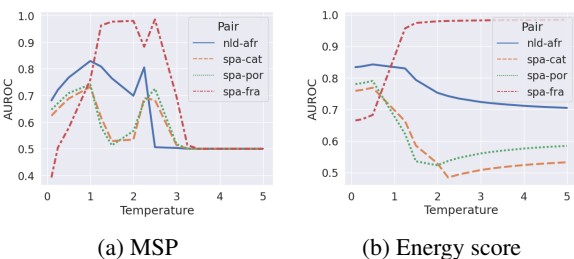

(a) MSP  (b) Energy score

Figure 4: Impact of the temperature used to compute the energy ($a_E$) and MSP ($a_{MSP}$) OOD scores in terms of AUROC.

## F.4 ROC AUC curves

### F.4.1 Language shifts

In Fig. 7 and Fig. 8 we present the ROC-AUC curves of our different detectors for language shifts in translation.

### F.4.2 Domain shifts

In Fig. 9 and Fig. 10 we present the ROC-AUC curves of our different detectors for topic shifts in translation.

### F.4.3 Dialog shifts

In Fig. 11 and Fig. 12 we present the ROC-AUC curves of our different detectors for topic shifts in a dialog setting.

## G NTM performance

Surprisingly we show that common OOD detectors tend to exclude samples that the model well handles and keep some that are not leading to decreasing overall performance in terms of translation metrics. Moreover, it seems this phenomenon is more dominant in reference-based detectors. We show that our uncertainty-based detectors mostly avoid that downfall and provide good OOD detection and improved translation performances.

### G.1 Absolute performances

It is clear (somewhat expected) that NMT models do not perform as well on OOD data as we can see in Tab. 19b. However, we find that our OOD detectors are able to remove most of the worst-case samples and keep enough well-translated samples so that with correct filtering our method actually allows the model to achieve somewhat acceptable BLEU scores.

### G.2 Gains

In Tab. 20 we give the detailed gain in translation performance based on the BLEU score.

### G.3 Choice of threshold

We believe that the choice of the threshold for OOD detection should not require OOD samples because we do not want to assume we have access to all kind

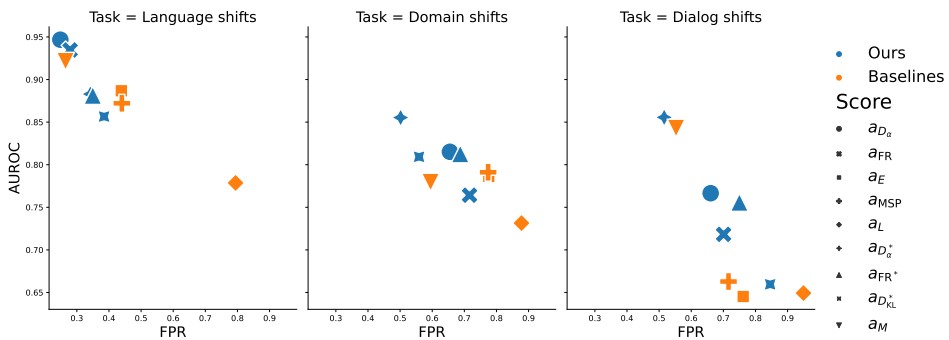

Figure 6: **Trade-offs** between AUROC and FPR ↓ for each tasks and metrics

Table 18: Detailed performance results of our OOD detectors on dialog shift against the Multi WOZ dataset as reference set.

| Scenario | | Score | AUPR-IN | AUPR-OUT | AUROC | ERR | f1 | FPR | precision | recall |
|---|---|---|---|---|---|---|---|---|---|---|
| $s_0$ | Ours | $a_{D_\alpha}$ | **0.87** | **0.87** | **0.87** | **0.31** | 0.78 | **0.56** | 0.79 | 0.77 |
| | | $a_{FR}$ | 0.73 | 0.81 | 0.79 | 0.34 | 0.67 | 0.63 | 0.75 | 0.60 |
| | | $a_E$ | 0.75 | 0.75 | 0.62 | 0.40 | 0.67 | 0.75 | 0.53 | 1.00 |
| | Baselines | $a_L$ | 0.81 | 0.79 | 0.82 | 0.39 | 0.76 | 0.72 | 0.75 | 0.77 |
| | | $a_{MSP}$ | 0.53 | 0.64 | 0.57 | 0.42 | 0.32 | 0.78 | 0.53 | 0.22 |
| | | $a_{D_\alpha^*}$ | 0.69 | 0.69 | 0.72 | 0.43 | 0.60 | 0.81 | 0.72 | 0.52 |
| | | $a_{D_{KL}^*}$ | 0.66 | 0.68 | 0.70 | 0.44 | 0.56 | 0.82 | 0.70 | 0.47 |
| $s_1$ | Ours | $a_{D_\alpha^{mean}}$ | 0.62 | 0.65 | 0.65 | 0.44 | 0.47 | 0.83 | 0.65 | 0.37 |
| | | $a_{D_{KL}^{mean}}$ | 0.64 | 0.65 | 0.67 | 0.46 | 0.51 | 0.86 | 0.67 | 0.41 |
| | | $a_{FR^*}$ | 0.69 | 0.69 | 0.72 | 0.43 | 0.60 | 0.82 | 0.71 | 0.51 |
| | | $a_{FR^{mean}}$ | 0.55 | 0.57 | 0.57 | 0.48 | 0.37 | 0.91 | 0.58 | 0.27 |
| | Baselines | $a_C$ | 0.88 | 0.89 | 0.90 | 0.25 | 0.75 | 0.46 | 0.86 | 0.67 |
| | | $a_M$ | 0.87 | 0.84 | 0.87 | 0.33 | 0.80 | 0.61 | 0.80 | 0.80 |

(a) dailydialog-default

| Scenario | | Score | AUPR-IN | AUPR-OUT | AUROC | ERR | f1 | FPR | precision | recall |
|---|---|---|---|---|---|---|---|---|---|---|
| $s_0$ | Ours | $a_{D_\alpha}$ | 0.52 | **0.87** | **0.72** | **0.52** | 0.52 | **0.69** | 0.50 | 0.54 |
| | | $a_{FR}$ | 0.41 | 0.86 | 0.69 | 0.55 | 0.37 | 0.73 | 0.39 | 0.35 |
| | | $a_E$ | **0.63** | **0.87** | 0.63 | 0.56 | 0.43 | 0.74 | 0.28 | 1.00 |
| | Baselines | $a_L$ | 0.40 | 0.66 | 0.47 | 0.73 | 0.43 | 1.00 | 0.27 | 1.00 |
| | | $a_{MSP}$ | 0.36 | 0.86 | 0.67 | **0.52** | 0.31 | **0.69** | 0.34 | 0.28 |
| | | $a_{D_\alpha^*}$ | 0.69 | 0.93 | 0.85 | 0.40 | 0.63 | 0.53 | 0.56 | 0.75 |
| | | $a_{D_{KL}^*}$ | 0.42 | 0.84 | 0.67 | 0.63 | 0.43 | 0.85 | 0.44 | 0.43 |
| $s_1$ | Ours | $a_{D_\alpha^{mean}}$ | 0.39 | 0.82 | 0.64 | 0.63 | 0.38 | 0.85 | 0.40 | 0.36 |
| | | $a_{D_{KL}^{mean}}$ | 0.39 | 0.81 | 0.64 | 0.67 | 0.40 | 0.90 | 0.41 | 0.38 |
| | | $a_{FR^*}$ | 0.53 | 0.88 | 0.75 | 0.58 | 0.54 | 0.77 | 0.51 | 0.59 |
| | | $a_{FR^{mean}}$ | 0.31 | 0.80 | 0.58 | 0.66 | 0.29 | 0.88 | 0.33 | 0.26 |
| | Baselines | $a_C$ | 0.54 | 0.89 | 0.70 | 0.74 | 0.59 | 0.91 | 0.55 | 0.64 |
| | | $a_M$ | 0.53 | 0.87 | 0.74 | 0.73 | 0.56 | 0.99 | 0.53 | 0.60 |

(b) silicone-melds

| Scenario | | Score | AUPR-IN | AUPR-OUT | AUROC | ERR | f1 | FPR | precision | recall |
|---|---|---|---|---|---|---|---|---|---|---|
| $s_0$ | Ours | $a_{D_\alpha}$ | 0.52 | **0.87** | **0.72** | **0.52** | 0.52 | **0.69** | 0.50 | 0.54 |
| | | $a_{FR}$ | 0.41 | 0.86 | 0.69 | 0.55 | 0.37 | 0.73 | 0.39 | 0.35 |
| | | $a_E$ | **0.63** | **0.87** | 0.63 | 0.56 | 0.43 | 0.74 | 0.28 | 1.00 |
| | Baselines | $a_L$ | 0.40 | 0.66 | 0.47 | 0.73 | 0.43 | 1.00 | 0.27 | 1.00 |
| | | $a_{MSP}$ | 0.36 | 0.86 | 0.67 | **0.52** | 0.31 | **0.69** | 0.34 | 0.28 |
| | | $a_{D_\alpha^*}$ | 0.69 | 0.93 | 0.85 | 0.40 | 0.63 | 0.53 | 0.56 | 0.75 |
| | | $a_{D_{KL}^*}$ | 0.42 | 0.84 | 0.67 | 0.63 | 0.43 | 0.85 | 0.44 | 0.43 |
| $s_1$ | Ours | $a_{D_\alpha^{mean}}$ | 0.39 | 0.83 | 0.65 | 0.63 | 0.39 | 0.84 | 0.41 | 0.38 |
| | | $a_{D_{KL}^{mean}}$ | 0.40 | 0.82 | 0.65 | 0.65 | 0.40 | 0.88 | 0.41 | 0.38 |
| | | $a_{FR^*}$ | 0.53 | 0.88 | 0.75 | 0.58 | 0.54 | 0.77 | 0.51 | 0.59 |
| | | $a_{FR^{mean}}$ | 0.32 | 0.79 | 0.59 | 0.68 | 0.32 | 0.91 | 0.35 | 0.29 |
| | Baselines | $a_C$ | 0.54 | 0.89 | 0.70 | 0.74 | 0.59 | 0.91 | 0.55 | 0.64 |
| | | $a_M$ | 0.53 | 0.87 | 0.74 | 0.73 | 0.56 | 0.99 | 0.53 | 0.60 |

(c) silicone-melde

| Scenario | | Score | AUPR-IN | AUPR-OUT | AUROC | ERR | f1 | FPR | precision | recall |
|---|---|---|---|---|---|---|---|---|---|---|
| $s_0$ | Ours | $a_{D_\alpha}$ | 0.79 | **0.80** | 0.81 | **0.36** | 0.73 | **0.68** | 0.78 | 0.69 |
| | | $a_{FR}$ | 0.68 | 0.73 | 0.70 | 0.39 | 0.49 | 0.74 | 0.66 | 0.39 |
| | | $a_E$ | 0.75 | 0.75 | 0.64 | 0.45 | 0.67 | 0.84 | 0.61 | 1.00 |
| | Baselines | $a_L$ | **0.91** | 0.73 | **0.85** | 0.49 | 0.78 | 0.94 | 0.76 | 0.81 |
| | | $a_{MSP}$ | 0.70 | 0.65 | 0.64 | 0.43 | 0.54 | 0.80 | 0.69 | 0.45 |
| | | $a_{D_\alpha^*}$ | 0.92 | 0.91 | 0.91 | 0.23 | 0.83 | 0.42 | 1.00 | 0.87 |
| | | $a_{D_{KL}^*}$ | 0.60 | 0.60 | 0.61 | 0.47 | 0.44 | 0.89 | 0.63 | 0.34 |
| $s_1$ | Ours | $a_{D_\alpha^{mean}}$ | 0.59 | 0.59 | 0.60 | 0.47 | 0.41 | 0.90 | 0.61 | 0.31 |
| | | $a_{D_{KL}^{mean}}$ | 0.59 | 0.58 | 0.59 | 0.48 | 0.41 | 0.90 | 0.61 | 0.31 |
| | | $a_{FR^*}$ | 0.75 | 0.74 | 0.75 | 0.41 | 0.65 | 0.77 | 0.74 | 0.58 |
| | | $a_{FR^{mean}}$ | 0.56 | 0.59 | 0.59 | 0.47 | 0.39 | 0.89 | 0.59 | 0.29 |
| | Baselines | $a_C$ | 0.87 | 0.87 | 0.88 | 0.15 | 0.80 | 0.21 | 0.80 | 0.84 |
| | | $a_M$ | 0.85 | 0.93 | 0.91 | 0.10 | 0.83 | 0.21 | 0.81 | 0.85 |

(d) silicone-dydae

| Scenario | | Score | AUPR-IN | AUPR-OUT | AUROC | ERR | f1 | FPR | precision | recall |
|---|---|---|---|---|---|---|---|---|---|---|
| $s_0$ | Ours | $a_{D_\alpha}$ | 0.71 | 0.74 | **0.72** | **0.36** | 0.63 | 0.67 | 0.73 | 0.55 |
| | | $a_{FR}$ | 0.68 | 0.73 | **0.72** | **0.36** | 0.57 | 0.67 | 0.70 | 0.48 |
| | | $a_E$ | **0.75** | **0.75** | 0.66 | 0.39 | 0.67 | 0.74 | 0.53 | 1.00 |
| | Baselines | $a_L$ | 0.69 | 0.48 | 0.57 | 0.50 | 0.67 | 1.00 | 0.50 | 1.00 |
| | | $a_{MSP}$ | 0.61 | 0.74 | 0.71 | **0.36** | 0.00 | **0.66** | 0.00 | 0.00 |
| | | $a_{D_\alpha^*}$ | 0.88 | 0.89 | 0.88 | 0.24 | 0.80 | 0.44 | 0.79 | 0.82 |
| | | $a_{D_{KL}^*}$ | 0.68 | 0.70 | 0.70 | 0.42 | 0.56 | 0.80 | 0.70 | 0.47 |
| $s_1$ | Ours | $a_{D_\alpha^{mean}}$ | 0.64 | 0.67 | 0.67 | 0.43 | 0.49 | 0.82 | 0.66 | 0.39 |
| | | $a_{D_{KL}^{mean}}$ | 0.65 | 0.66 | 0.67 | 0.44 | 0.51 | 0.84 | 0.67 | 0.41 |
| | | $a_{FR^*}$ | 0.79 | 0.80 | 0.80 | 0.36 | 0.71 | 0.68 | 0.76 | 0.66 |
| | | $a_{FR^{mean}}$ | 0.57 | 0.59 | 0.59 | 0.47 | 0.38 | 0.89 | 0.58 | 0.28 |
| | Baselines | $a_C$ | 0.87 | 0.87 | 0.88 | 0.15 | 0.80 | 0.21 | 0.80 | 0.84 |
| | | $a_M$ | 0.87 | 0.95 | 0.93 | 0.10 | 0.86 | 0.21 | 0.82 | 0.90 |

(e) silicone-swda

| Scenario | | Score | AUPR-IN | AUPR-OUT | AUROC | ERR | f1 | FPR | precision | recall |
|---|---|---|---|---|---|---|---|---|---|---|
| $s_0$ | Ours | $a_{D_\alpha}$ | 0.79 | **0.80** | 0.81 | **0.36** | 0.73 | **0.68** | 0.78 | 0.69 |
| | | $a_{FR}$ | 0.68 | 0.73 | 0.70 | 0.39 | 0.49 | 0.74 | 0.66 | 0.39 |
| | | $a_E$ | 0.75 | 0.75 | 0.64 | 0.45 | 0.67 | 0.84 | 0.61 | 1.00 |
| | Baselines | $a_L$ | **0.91** | 0.73 | **0.85** | 0.49 | 0.78 | 0.94 | 0.76 | 0.81 |
| | | $a_{MSP}$ | 0.70 | 0.65 | 0.64 | 0.43 | 0.54 | 0.80 | 0.69 | 0.45 |
| | | $a_{D_\alpha^*}$ | 0.92 | 0.91 | 0.91 | 0.23 | 0.83 | 0.42 | 1.00 | 0.87 |
| | | $a_{D_{KL}^*}$ | 0.60 | 0.60 | 0.61 | 0.47 | 0.44 | 0.89 | 0.63 | 0.34 |
| $s_1$ | Ours | $a_{D_\alpha^{mean}}$ | 0.59 | 0.59 | 0.60 | 0.47 | 0.41 | 0.90 | 0.61 | 0.31 |
| | | $a_{D_{KL}^{mean}}$ | 0.58 | 0.57 | 0.59 | 0.49 | 0.41 | 0.92 | 0.61 | 0.31 |
| | | $a_{FR^*}$ | 0.75 | 0.74 | 0.75 | 0.41 | 0.65 | 0.77 | 0.74 | 0.58 |
| | | $a_{FR^{mean}}$ | 0.55 | 0.58 | 0.58 | 0.47 | 0.38 | 0.89 | 0.58 | 0.28 |
| | Baselines | $a_C$ | 0.84 | 0.87 | 0.87 | 0.13 | 0.81 | 0.22 | 0.81 | 0.84 |
| | | $a_M$ | 0.85 | 0.93 | 0.91 | 0.10 | 0.83 | 0.21 | 0.81 | 0.85 |

(f) silicone-dydada

| Scenario | | Score | AUPR-IN | AUPR-OUT | AUROC | ERR | f1 | FPR | precision | recall |
|---|---|---|---|---|---|---|---|---|---|---|
| $s_0$ | Ours | $a_{D_\alpha}$ | 0.42 | **0.91** | **0.72** | 0.57 | 0.45 | **0.70** | 0.40 | 0.50 |
| | | $a_{FR}$ | 0.41 | 0.90 | **0.72** | 0.57 | 0.38 | 0.71 | 0.35 | 0.41 |
| | | $a_E$ | **0.61** | 0.89 | 0.63 | 0.61 | 0.35 | 0.76 | 0.23 | 1.00 |
| | Baselines | $a_L$ | 0.53 | 0.76 | 0.58 | 0.80 | 0.40 | 1.00 | 0.34 | 0.48 |
| | | $a_{MSP}$ | 0.35 | 0.90 | 0.68 | **0.56** | 0.33 | **0.70** | 0.32 | 0.35 |
| | | $a_{D_\alpha^*}$ | 0.67 | 0.95 | 0.86 | 0.44 | 0.58 | 0.55 | 0.48 | 0.77 |
| | | $a_{D_{KL}^*}$ | 0.29 | 0.85 | 0.63 | 0.72 | 0.34 | 0.90 | 0.32 | 0.36 |
| $s_1$ | Ours | $a_{D_\alpha^{mean}}$ | 0.27 | 0.85 | 0.61 | 0.72 | 0.30 | 0.89 | 0.29 | 0.31 |
| | | $a_{D_{KL}^{mean}}$ | 0.28 | 0.85 | 0.61 | 0.71 | 0.30 | 0.89 | 0.29 | 0.31 |
| | | $a_{FR^*}$ | 0.41 | 0.90 | 0.73 | 0.61 | 0.46 | 0.77 | 0.41 | 0.52 |
| | Baselines | $a_C$ | 0.45 | 0.92 | 0.75 | 0.78 | 0.53 | 0.97 | 0.50 | 0.56 |
| | | $a_M$ | 0.42 | 0.89 | 0.70 | 0.78 | 0.47 | 0.99 | 0.42 | 0.54 |

(g) silicone-iemocap

| Scenario | | Score | AUPR-IN | AUPR-OUT | AUROC | ERR | f1 | FPR | precision | recall |
|---|---|---|---|---|---|---|---|---|---|---|
| $s_0$ | Ours | $a_{D_\alpha}$ | 0.71 | **0.77** | **0.75** | 0.33 | 0.60 | 0.60 | 0.72 | 0.52 |
| | | $a_{FR}$ | 0.69 | 0.74 | 0.74 | 0.36 | 0.58 | 0.67 | 0.71 | 0.49 |
| | | $a_E$ | **0.75** | 0.75 | 0.70 | 0.37 | 0.67 | 0.68 | 0.60 | 1.00 |
| | Baselines | $a_L$ | 0.72 | 0.48 | 0.57 | 0.50 | 0.67 | 1.00 | 0.50 | 1.00 |
| | | $a_{MSP}$ | 0.62 | 0.76 | 0.73 | **0.32** | 0.00 | **0.59** | 0.00 | 0.00 |
| | | $a_{D_\alpha^*}$ | 0.86 | 0.88 | 0.87 | 0.23 | 0.79 | 0.42 | 0.79 | 0.80 |
| | | $a_{D_{KL}^*}$ | 0.66 | 0.71 | 0.70 | 0.41 | 0.52 | 0.77 | 0.68 | 0.42 |
| $s_1$ | Ours | $a_{D_\alpha^{mean}}$ | 0.63 | 0.68 | 0.66 | 0.42 | 0.45 | 0.79 | 0.64 | 0.35 |
| | | $a_{D_{KL}^{mean}}$ | 0.63 | 0.65 | 0.66 | 0.44 | 0.48 | 0.84 | 0.66 | 0.38 |
| | | $a_{FR^*}$ | 0.77 | 0.80 | 0.79 | 0.36 | 0.69 | 0.66 | 0.75 | 0.63 |
| | | $a_{FR^{mean}}$ | 0.57 | 0.59 | 0.59 | 0.47 | 0.40 | 0.90 | 0.60 | 0.30 |
| | Baselines | $a_C$ | 0.87 | 0.92 | 0.94 | 0.13 | 0.89 | 0.22 | 0.82 | 0.65 |
| | | $a_M$ | 0.88 | 0.96 | 0.94 | 0.10 | 0.88 | 0.21 | 0.82 | 0.94 |

(h) silicone-mrda

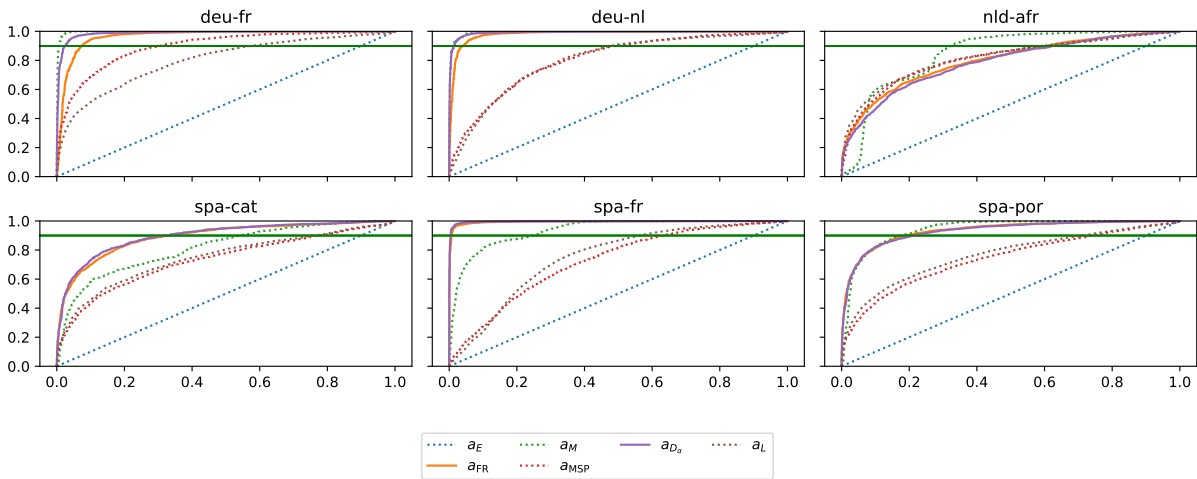

Figure 7: ROCAUC curves for our uncertainty-based metrics compared to common baselines for language shift detection. Baselines are represented in dashed lines.

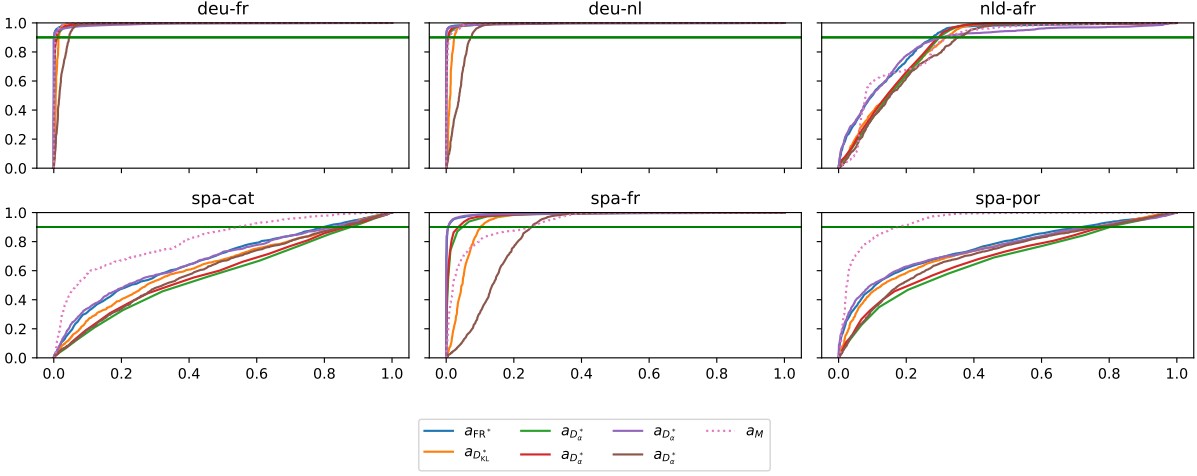

Figure 8: **ROC-AUC curves** for our reference-based metrics compared to common baselines for language shift detection. Baselines are represented in dashed lines.

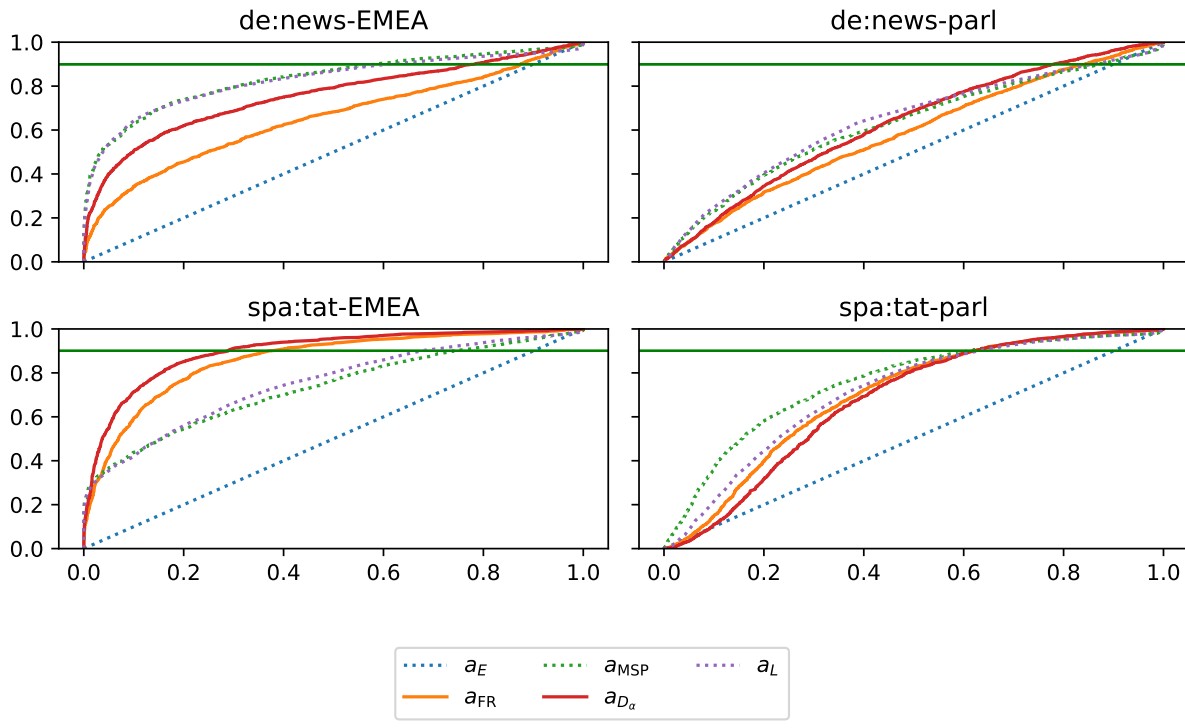

Figure 9: **ROC-AUC curves** for our uncertainty-based metrics compared to common baselines for domain shift detection. baselines are represented in dashed lines.

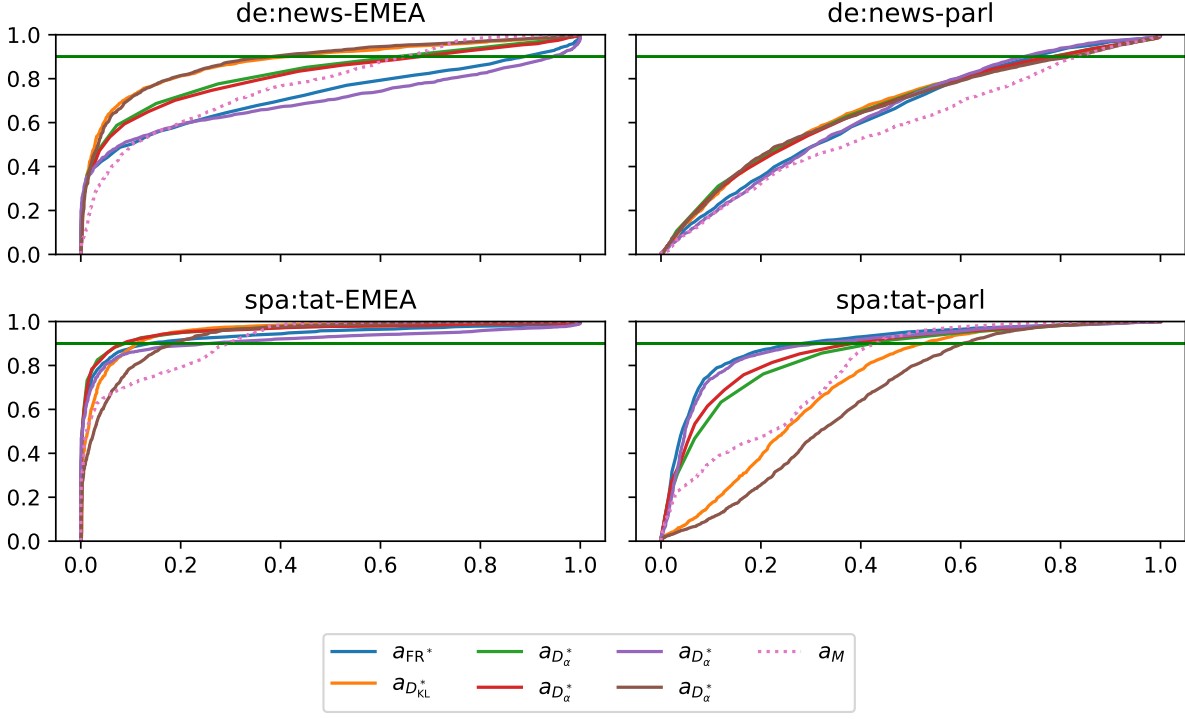

Figure 10: ROC-AUC curves for our reference-based metrics compared to common baselines for domain shift detection. baselines are represented in dashed lines.

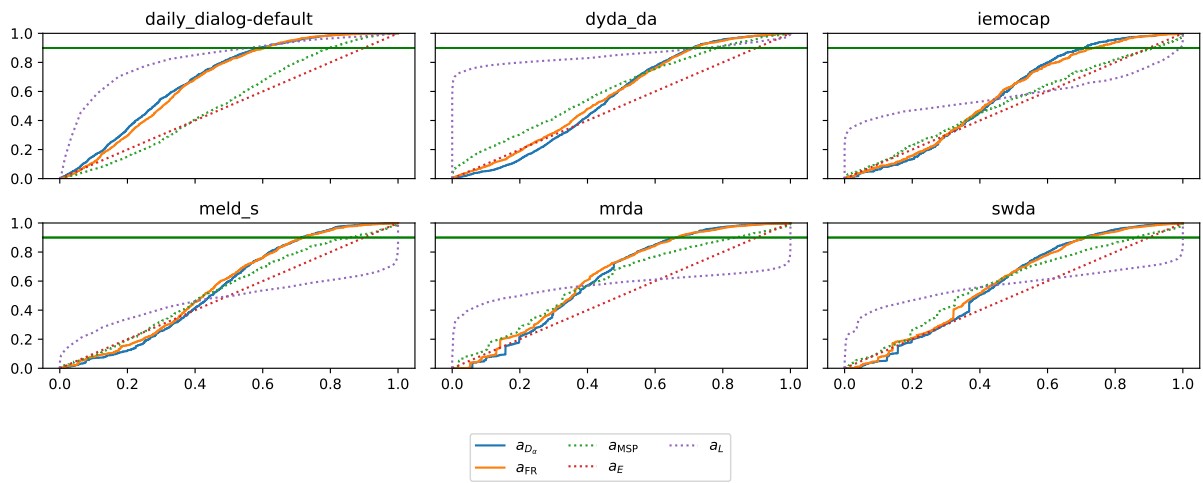

Figure 11: **ROC-AUC curves** for our uncertainty-based metrics compared to common baselines for dialog shift detection. baselines are represented in dashed lines.

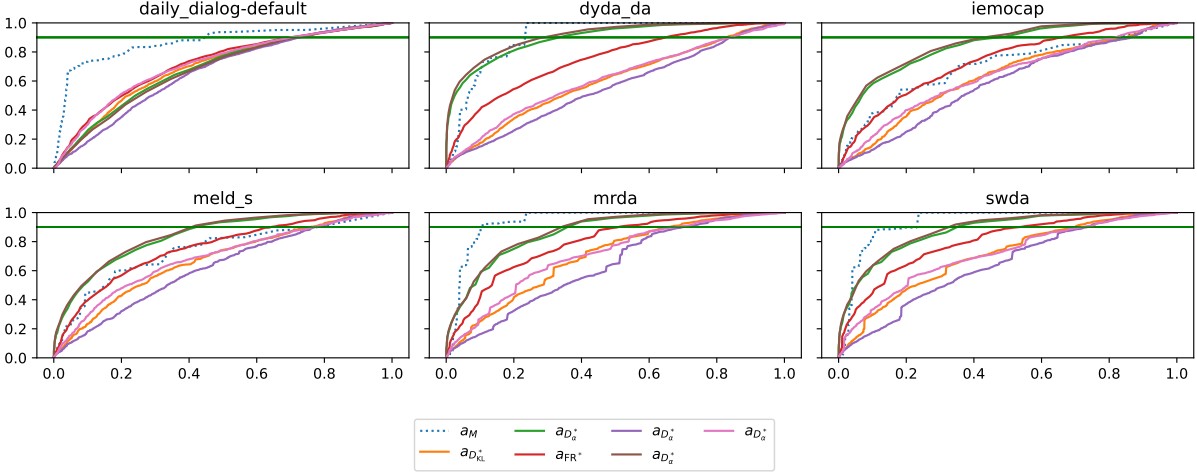

Figure 12: **ROC-AUC curves** for our reference-based metrics compared to common baselines for dialog shift detection. baselines are represented in dashed lines.

| Scenario | | Score | spa-cat | spa-por | nld-afr | spa:tat-parl | deu:news-parl | spa:tat-EMEA | deu:news-EMEA |
|---|---|---|---|---|---|---|---|---|---|
| | | ✗ | 59.63 | 59.63 | 62.38 | 59.63 | 34.07 | 59.63 | 34.07 |
| $\$_0$ | Ours | $a_{D_\alpha}$ | 63.82 | 62.41 | 67.77 | 64.78 | 36.68 | 63.82 | 36.69 |
| | | $a_{FR}$ | 62.41 | 62.41 | 65.89 | 63.54 | 36.10 | 63.54 | 36.19 |
| | Baselines | $a_E$ | 62.99 | 62.99 | 65.80 | 62.99 | 35.87 | 62.99 | 35.87 |
| | | $a_L$ | 64.52 | 64.52 | 67.78 | 64.52 | 36.72 | 64.52 | 36.72 |
| | | $a_{MSP}$ | 58.15 | 58.15 | 60.31 | 58.15 | 33.00 | 58.15 | 33.00 |
| | | $a_{CQE}$ | 60.36 | 60.36 | 63.19 | 60.36 | 35.68 | 60.36 | 35.68 |
| $\$_1$ | Ours | $a_{D_\alpha^*}$ | 59.99 | 59.99 | 62.82 | 59.99 | 33.82 | 59.99 | 33.82 |
| | | $a_{FR^*}$ | 59.98 | 59.98 | 62.83 | 59.98 | 33.78 | 59.98 | 33.78 |
| | Baselines | $a_C$ | 60.80 | 60.80 | 62.67 | 60.80 | 34.48 | 60.80 | 34.48 |
| | | $a_M$ | 59.63 | 59.63 | 62.61 | 59.63 | 33.85 | 59.63 | 33.85 |

(a) IN

| Scenario | | Score | spa-cat | spa-por | nld-afr | spa:tat-parl | deu:news-parl | spa:tat-EMEA | deu:news-EMEA |
|---|---|---|---|---|---|---|---|---|---|
| | | ✗ | 15.73 | 15.40 | 23.79 | 33.17 | 28.36 | 59.38 | 52.16 |
| $\$_0$ | Ours | $a_{D_\alpha}$ | 22.06 | 38.04 | 36.51 | 41.80 | 33.64 | 64.11 | 54.60 |
| | | $a_{FR}$ | 26.41 | 37.74 | 33.86 | 38.17 | 32.10 | 62.29 | 55.02 |
| | Baselines | $a_E$ | 14.44 | 12.97 | 27.64 | 35.11 | 31.97 | 60.52 | 54.48 |
| | | $a_L$ | 16.62 | 14.70 | 35.90 | 40.23 | 34.18 | 61.32 | 54.41 |
| | | $a_{MSP}$ | 17.36 | 18.97 | 23.78 | 30.81 | 26.46 | 41.51 | 34.64 |
| | | $a_{CQE}$ | 17.61 | 16.70 | 29.02 | 33.18 | 29.36 | 59.66 | 54.41 |
| $\$_1$ | Ours | $a_{D_\alpha^*}$ | 20.16 | 23.33 | 30.74 | 34.16 | 27.38 | 44.45 | 43.62 |
| | | $a_{FR^*}$ | 20.15 | 23.35 | 30.61 | 34.16 | 27.39 | 44.45 | 43.61 |
| | Baselines | $a_C$ | 16.84 | 16.72 | 23.67 | 33.32 | 29.11 | 59.55 | 52.19 |
| | | $a_M$ | 16.76 | 19.37 | 25.64 | 30.83 | 27.22 | 59.73 | 54.71 |

(b) OOD

| Scenario | | Score | spa-cat | spa-por | nld-afr | spa:tat-parl | deu:news-parl | spa:tat-EMEA | deu:news-EMEA |
|---|---|---|---|---|---|---|---|---|---|
| | | ✗ | 50.25 | 44.89 | 55.20 | 50.81 | 31.21 | 59.55 | 43.12 |
| $\$_0$ | Ours | $a_{D_\alpha}$ | 61.50 | 61.35 | 65.66 | 59.61 | 35.35 | 63.91 | 46.32 |
| | | $a_{FR}$ | 60.80 | 61.35 | 63.40 | 56.70 | 34.34 | 63.10 | 46.11 |
| | Baselines | $a_E$ | 54.86 | 48.72 | 61.99 | 53.66 | 34.19 | 62.06 | 45.78 |
| | | $a_L$ | 58.75 | 54.92 | 65.56 | 58.32 | 35.64 | 63.35 | 46.22 |
| | | $a_{MSP}$ | 51.26 | 48.32 | 52.96 | 48.65 | 29.50 | 56.86 | 33.37 |
| | | $a_{CQE}$ | 51.69 | 45.67 | 57.53 | 49.91 | 32.88 | 60.09 | 44.38 |
| $\$_1$ | Ours | $a_{D_\alpha^*}$ | 54.33 | 53.38 | 59.22 | 58.34 | 30.37 | 59.52 | 36.91 |
| | | $a_{FR^*}$ | 54.33 | 53.39 | 59.18 | 58.34 | 30.37 | 59.52 | 36.89 |
| | Baselines | $a_C$ | 53.27 | 48.88 | 55.85 | 50.62 | 31.77 | 60.34 | 43.84 |
| | | $a_M$ | 53.55 | 53.99 | 58.40 | 53.80 | 30.42 | 59.63 | 37.86 |

(c) ALL

Table 19: Absolue translation performances in terms of BLEU on the different subset (IN, OOD, ALL) of each dataset of our translation OOD performance benchmark.

**(a) IN**

| Scenario | | Score | spa-cat | spa-por | nld-afr | spa:tat-parl | deu:news-parl | spa:tat-EMEA | deu:news-EMEA |
|---|---|---|---|---|---|---|---|---|---|
| $S_0$ | Ours | $a_{D_\alpha}$ | +4.19 | +2.78 | +5.38 | +5.15 | +2.61 | +4.19 | +2.62 |
| | | $a_{\text{FR}}$ | +2.78 | +2.78 | +3.50 | +3.91 | +2.03 | +3.91 | +2.11 |
| | Baselines | $a_E$ | +3.36 | +3.36 | +3.42 | +3.36 | +1.79 | +3.36 | +1.79 |
| | | $a_L$ | +4.89 | +4.89 | +5.40 | +4.89 | +2.65 | +4.89 | +2.65 |
| | | $a_{\text{MSP}}$ | -1.48 | -1.48 | -2.07 | -1.48 | -1.07 | -1.48 | -1.07 |
| | | $a_{\text{CQE}}$ | +0.73 | +0.73 | +0.80 | +0.73 | +1.61 | +0.73 | +1.61 |
| $S_1$ | Ours | $a_{D_\alpha^*}$ | +0.36 | +0.36 | +0.44 | +0.36 | -0.26 | +0.36 | -0.26 |
| | | $a_{\text{FR}^*}$ | +0.35 | +0.35 | +0.45 | +0.35 | -0.29 | +0.35 | -0.29 |
| | Baselines | $a_C$ | +1.17 | +1.17 | +0.29 | +1.17 | +0.40 | +1.17 | +0.40 |
| | | $a_M$ | -0.00 | -0.00 | +0.23 | -0.00 | -0.22 | -0.00 | -0.22 |

(a) IN

**(b) OOD**

| Scenario | | Score | spa-cat | spa-por | nld-afr | spa:tat-parl | deu:news-parl | spa:tat-EMEA | deu:news-EMEA |
|---|---|---|---|---|---|---|---|---|---|
| $S_0$ | Ours | $a_{D_\alpha}$ | +6.33 | +22.64 | +12.72 | +8.62 | +5.28 | +4.74 | +2.44 |
| | | $a_{\text{FR}}$ | +10.68 | +22.34 | +10.07 | +5.00 | +3.75 | +2.91 | +2.85 |
| | Baselines | $a_E$ | -1.29 | -2.43 | +3.85 | +1.94 | +3.61 | +1.14 | +2.31 |
| | | $a_L$ | +0.89 | -0.70 | +12.12 | +7.06 | +5.82 | +1.94 | +2.24 |
| | | $a_{\text{MSP}}$ | +1.63 | +3.57 | -0.00 | -2.36 | -1.90 | -17.87 | -17.52 |
| | | $a_{\text{CQE}}$ | +1.88 | +1.30 | +5.23 | +0.01 | +1.00 | +0.29 | +2.25 |
| $S_1$ | Ours | $a_{D_\alpha^*}$ | +4.43 | +7.93 | +6.95 | +0.98 | -0.98 | -14.92 | -8.54 |
| | | $a_{\text{FR}^*}$ | +4.42 | +7.95 | +6.82 | +0.98 | -0.97 | -14.92 | -8.56 |
| | Baselines | $a_C$ | +1.11 | +1.32 | -0.12 | +0.14 | +0.75 | +0.17 | +0.02 |
| | | $a_M$ | +1.03 | +3.97 | +1.86 | -2.34 | -1.14 | +0.36 | +2.55 |

(b) OOD

**(c) ALL**

| Scenario | | Score | spa-cat | spa-por | nld-afr | spa:tat-parl | deu:news-parl | spa:tat-EMEA | deu:news-EMEA |
|---|---|---|---|---|---|---|---|---|---|
| $S_0$ | Ours | $a_{D_\alpha}$ | +11.24 | +16.46 | +10.46 | +8.80 | +4.14 | +4.37 | +3.20 |
| | | $a_{\text{FR}}$ | +10.55 | +16.46 | +8.20 | +5.89 | +3.13 | +3.56 | +2.99 |
| | Baselines | $a_E$ | +4.61 | +3.83 | +6.80 | +2.85 | +2.98 | +2.52 | +2.66 |
| | | $a_L$ | +8.50 | +10.04 | +10.36 | +7.51 | +4.42 | +3.80 | +3.10 |
| | | $a_{\text{MSP}}$ | +1.01 | +3.43 | -2.24 | -2.16 | -1.72 | -2.68 | -9.74 |
| | | $a_{\text{CQE}}$ | +1.44 | +0.78 | +2.33 | -0.90 | +1.67 | +0.55 | +1.26 |
| $S_1$ | Ours | $a_{D_\alpha^*}$ | +4.07 | +8.49 | +4.03 | +7.53 | -0.85 | -0.02 | -6.20 |
| | | $a_{\text{FR}^*}$ | +4.08 | +8.50 | +3.98 | +7.52 | -0.85 | -0.03 | -6.23 |
| | Baselines | $a_C$ | +3.02 | +3.99 | +0.65 | -0.19 | +0.56 | +0.80 | +0.72 |
| | | $a_M$ | +3.30 | +9.10 | +3.20 | +2.99 | -0.79 | +0.08 | -5.26 |

(c) ALL

Table 20: Detailed impact of the OOD filtering on the different subset for each task.

| Scenario | | Dataset Score | spa-cat | | | spa-por | | | nld-afr | | | spa:tat-parl | | | deu:news-parl | | | spa:tat-EMEA | | | deu:news-EMEA | | |
|---|---|---|---|---|---|---|---|---|---|---|---|---|---|---|---|---|---|---|---|---|---|---|---|
| | | | ALL | IN | OOD | ALL | IN | OOD | ALL | IN | OOD | ALL | IN | OOD | ALL | IN | OOD | ALL | IN | OOD | ALL | IN | OOD |
| $S_0$ | Ours | $a_{D_\alpha}$ | 32% | 18% | 82% | 44% | 20% | 93% | 30% | 20% | 75% | 31% | 20% | 54% | 29% | 20% | 38% | 19% | 18% | 20% | 14% | 20% | 7% |
| | | $a_{\mathrm{FR}}$ | 34% | 20% | 86% | 44% | 20% | 93% | 26% | 17% | 69% | 24% | 17% | 39% | 28% | 20% | 37% | 16% | 17% | 12% | 15% | 19% | 10% |
| | Baselines | $a_E$ | 24% | 20% | 41% | 25% | 20% | 36% | 28% | 20% | 61% | 20% | 20% | 20% | 30% | 20% | 40% | 15% | 20% | 4% | 14% | 20% | 9% |
| | | $a_L$ | 27% | 19% | 59% | 33% | 19% | 61% | 30% | 20% | 74% | 27% | 19% | 44% | 30% | 19% | 40% | 14% | 19% | 6% | 13% | 19% | 6% |
| | | $a_{\mathrm{MSP}}$ | 22% | 18% | 38% | 27% | 18% | 45% | 17% | 19% | 10% | 16% | 18% | 12% | 14% | 20% | 8% | 41% | 18% | 86% | 48% | 20% | 76% |
| | | $a_{\mathrm{CQE}}$ | 21% | 20% | 25% | 20% | 20% | 19% | 22% | 20% | 31% | 13% | 20% | 0% | 28% | 20% | 37% | 14% | 20% | 1% | 25% | 20% | 31% |
| $S_1$ | Ours | $a_{D^*_\alpha}$ | 26% | 19% | 51% | 34% | 19% | 65% | 26% | 20% | 56% | 43% | 19% | 89% | 12% | 19% | 6% | 45% | 19% | 95% | 40% | 19% | 62% |
| | | $a_{\mathrm{FR}^*}$ | 26% | 19% | 51% | 35% | 19% | 65% | 25% | 19% | 55% | 43% | 19% | 95% | 12% | 18% | 6% | 45% | 19% | 95% | 40% | 18% | 62% |
| | Baselines | $a_C$ | 21% | 17% | 37% | 24% | 17% | 38% | 17% | 16% | 22% | 12% | 11% | 2% | 10% | 11% | 10% | 12% | 17% | 2% | 6% | 11% | 0% |
| | | $a_M$ | 27% | 20% | 51% | 38% | 20% | 74% | 27% | 20% | 55% | 33% | 20% | 59% | 17% | 20% | 14% | 45% | 20% | 96% | 50% | 20% | 81% |

Table 21: Share of the datasets removed when taking $\gamma$ so that we keep 80% of the IN distribution.

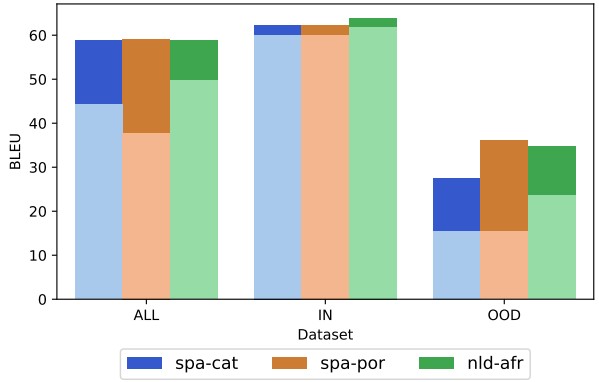

Figure 13: **Gain in translation performances** when filtering OOD samples with our method on different datasets and language pairs.

Table 22: **Correlation between OOD scores and translation metrics** BLEU and BERT-S on domain shifts datasets.

| | | | BERT-S | | | BLEU | | | COMET | | |
|---|---|---|---|---|---|---|---|---|---|---|---|
| | | Score | ALL | IN | OUT | ALL | IN | OUT | ALL | IN | OUT |
| $S_0$ | Ours | $a_{D_\alpha}$ | -0.31 | -0.25 | -0.18 | -0.19 | -0.22 | -0.09 | -0.29 | -0.29 | -0.17 |
| | | $a_{\mathrm{FR}}$ | -0.37 | -0.29 | -0.27 | -0.25 | -0.25 | -0.19 | -0.34 | -0.33 | -0.25 |
| | Bas. | $a_E$ | 0.16 | 0.25 | 0.33 | 0.22 | 0.20 | 0.39 | 0.21 | 0.26 | 0.33 |
| | | $a_L$ | 0.46 | 0.50 | 0.48 | 0.48 | 0.45 | 0.49 | 0.48 | 0.50 | 0.46 |
| | | $a_{\mathrm{MSP}}$ | 0.12 | 0.19 | 0.29 | 0.20 | 0.16 | 0.37 | 0.16 | 0.19 | 0.28 |
| | | $a_{\mathrm{CQE}}$ | -0.03 | 0.12 | 0.28 | -0.01 | 0.05 | 0.13 | 0.14 | 0.25 | 0.45 |
| $S_1$ | Ours | $a_{D^*_\alpha}$ | -0.24 | 0.01 | 0.10 | -0.08 | -0.00 | 0.19 | -0.15 | -0.02 | 0.13 |
| | | $a_{\mathrm{FR}^*}$ | -0.21 | 0.02 | 0.08 | -0.08 | 0.01 | 0.15 | -0.14 | -0.02 | 0.12 |
| | Bas. | $a_M$ | -0.20 | -0.02 | 0.00 | -0.04 | 0.00 | 0.09 | -0.13 | -0.02 | -0.00 |

of different OOD samples that might occur. Therefore we choose to fix the False Positive Rate of our detector by constraining the amount of known IN distribution samples that are classified as OOD.

# H  Negative results

## H.1  Different aggregation of OOD metrics

Most of our detectors are initially classification OOD detectors that we adapted for text generation by averaging them over the generated sequences and using this aggregated score as a score for the whole sequence. We experimented with other aggregations such as the standard deviation or the min/max along the sequence. If the standard deviation gave relatively good results they were still less interesting that the naive average.

## H.2  Negentropy of bag of distributions

We introduced in Sec. 3.3 the bag of distributions as a way to aggregate a sequence of probability distribution and compare it to a set of reference using information projections Sec. 3.3. A natural idea would be to apply the Negentropy methods (Sec. 3.2) to these aggregated distributions.

More formally given a sequence of probability distribution $\mathcal{S}_\theta(\mathbf{x}) = \{p_\theta^T(\mathbf{x}, \hat{\mathbf{y}}_{\leqslant t})\}_{t=1}^n$ we would compute its bag of distributions:

$$\bar{p}_\theta(x) \triangleq \frac{1}{|y|} \sum_{t=1}^{|y|} p_\theta(x, y_{\leqslant t}) \qquad (7)$$

And then compute as novelty score:

$$J_D(\mathbf{p}) = D(\mathbf{p} \| \mathcal{U}). \qquad (8)$$

Further experiments have shown that this process was unable to discriminate OOD samples or improve performance translation. We suspect that the uncertainty at each step is key to capture the behavior of the language model and that this uncertainty information is lost when averaging probability distribution along the sequence.

Table 23: **Detailed impacts on NMT performance results per tasks (Domain- or Language-shifts)** of the different OOD detectors with a threshold defined to keep $99\%$ of the IN data. We present results on the different part of the data: IN data, OOD data and the combination of both, ALL. For each we report the absolute average BLEU score (Abs.), the average gains in BLEU (G.s.) compared to a setting without OOD filtering ($f_\theta$ only) and the share of the subset removed by the detector (R.Sh.).

| | | | Domain shifts | | | | | | | | | | Language shifts | | | | | | | | |
| | | | IN | | | OOD | | | ALL | | | IN | | | OOD | | | ALL | | |
| | | | Abs. | G.s | R.Sh | Abs. | G.s | R.Sh | Abs. | G.s | R.Sh | Abs. | G.s | R.Sh | Abs. | G.s | R.Sh | Abs. | G.s | R.Sh |
| | | $\times$ | 47.1 | +0.0 | 0.0% | 43.4 | +0.0 | 0.0% | 45.3 | +0.0 | 0.0% | 60.5 | +0.0 | 0.0% | 18.1 | +0.0 | 0.0% | 43.9 | +0.0 | 0.0% |
| $\mathfrak{s}_0$ | Ours | $a_{D_\alpha}$ | 47.3 | +0.2 | 1.0% | 44.2 | +0.8 | 5.5% | 45.8 | +0.5 | 3.2% | 60.7 | +0.2 | 1.0% | 21.8 | +3.7 | 34.5% | 49.2 | +5.4 | 14.6% |
| | | $a_{\mathrm{FR}}$ | 47.3 | +0.2 | 1.0% | 44.1 | +0.7 | 7.2% | 45.7 | +0.4 | 4.1% | 60.7 | +0.2 | 1.0% | 22.3 | +4.2 | 37.0% | 49.7 | +5.9 | 15.8% |
| | Bas. | $a_E$ | 47.3 | +0.2 | 1.0% | 44.0 | +0.5 | 1.9% | 45.6 | +0.4 | 1.4% | 60.9 | +0.4 | 1.0% | 18.7 | +0.6 | 17.6% | 46.0 | +2.1 | 7.3% |
| | | $a_L$ | 47.3 | +0.2 | 0.9% | 44.0 | +0.6 | 1.9% | 45.6 | +0.4 | 1.4% | 60.8 | +0.3 | 0.9% | 19.1 | +0.9 | 18.4% | 46.2 | +2.3 | 7.6% |
| | | $a_{\mathrm{MSP}}$ | 47.0 | -0.1 | 1.0% | 40.3 | -3.1 | 14.7% | 43.5 | -1.8 | 7.8% | 60.4 | -0.1 | 1.0% | 18.5 | +0.3 | 4.3% | 44.3 | +0.5 | 2.5% |
| $\mathfrak{s}_1$ | Ours | $a_{D_\alpha^*}$ | 47.0 | -0.1 | 0.9% | 40.3 | -3.1 | 26.5% | 43.9 | -1.4 | 13.7% | 60.5 | -0.0 | 0.9% | 19.3 | +1.1 | 10.5% | 45.3 | +1.4 | 4.8% |
| | | $a_{\mathrm{FR}^*}$ | 47.0 | -0.1 | 0.9% | 40.3 | -3.1 | 26.6% | 43.9 | -1.3 | 13.8% | 60.5 | -0.0 | 0.9% | 19.3 | +1.1 | 10.5% | 45.3 | +1.4 | 4.8% |
| | Bas. | $a_M$ | 47.0 | -0.1 | 1.0% | 41.6 | -1.8 | 18.1% | 44.6 | -0.7 | 9.6% | 60.5 | -0.0 | 1.0% | 18.4 | +0.3 | 12.1% | 45.3 | +1.4 | 5.9% |