# OpenReview forum: "RainProof: An Umbrella to Shield Text Generator from Out-Of-Distribution Data"
_EMNLP/2023/Conference — EMNLP 2023 Main_

### Official Review · Reviewer_j15V · 2023-08-02

**Soundness:** 3

**Excitement:**

3: Ambivalent: It has merits (e.g., it reports state-of-the-art results, the idea is nice), but there are key weaknesses (e.g., it describes incremental work), and it can significantly benefit from another round of revision. However, I won't object to accepting it if my co-reviewers champion it.

**Missing References:**

[1] Tim Z. Xiao et al. Wat zei je? Detecting Out-Of-Distribution Translations with Variational Transfromers. https://arxiv.org/abs/2006.08344

[2] Vazhentsev A. et al. Efficient Out-of-Domain Detection for Sequence to Sequence Models. in ACL 2023

[3] Guerreiro, Nuno M. et al. “Looking for a Needle in a Haystack: A Comprehensive Study of Hallucinations in Neural Machine Translation.” in EACL 2022.

[4] Podolskiy, A.V., et al. Revisiting Mahalanobis Distance for Transformer-Based Out-of-Domain Detection. in AAAI 2021

[5] Yiyou Sun, et al. Out-of-distribution detection with deep nearest neighbors. In ICML 2022.

[6] Mohammadreza Salehi et al. A Unified Survey on Anomaly, Novelty, Open-Set, and Out-of-Distribution Detection: solutions and future challenges. In TMLR 2022.

**Paper Topic And Main Contributions:**

The work addresses the problem of out-of-distribution (OOD) input detection for sequence generation tasks. The OOD detection task is crucial for the real-world application of models because the model’s performance could substantially drop on the OOD instances, hence, we shouldn’t provide such examples to the model, i.e. reject them. Authors are concerned with the translation and response generation tasks. They propose to use a new evaluation setting – Language out of distribution performance benchmark (LOFTER) that comprises machine translation (MT) and dialogues datasets. For MT, there are language shifts (e.g. Dutch model exposed to the Afrikaans) and domain shifts (e.g. EuroParl model exposed to the Tatoeba). For OOD detection, a new method is proposed (RAINPROOF) that can be used in two scenarios depending on whether in-distribution examples exist or not. Their method relies on the measurement of how the distribution of the generated instance differs from the proposal distribution (in an unsupervised regime) and from the closest distribution from in-distribution examples. The Renyi divergence and Fisher-Rao distance are used to measure the score.

The work is well-written, and the method is clearly described. There are a lot of different experiments presented in the paper and the Appendix. The suggested method outperforms the presented baselines on the proposed benchmark.

However, some questions arise and drawbacks exist.
First, the authors do not clearly describe why they needed to make a new benchmark and why they didn’t experiment with their approach on other benchmarks. For example, papers [1], [2], [3] propose different setups for NMT OOD detection. So, it seems that the paper lacks a comparison with the baselines and stronger models on other setups. Also, the authors do not consider the classification task with OOD examples like CLINC-150 which are commonly used, focusing mainly on generation tasks. However, it would be good to examine the approach to this setup.
Second, it is not clear why we choose to filter 80% of in-distribution samples. Do we choose it on some held-out data? Again, how do you choose hyper-parameters for the model if you do not use any OOD examples?
Third, why do we choose the uniform distribution as the reference, and what is the intuition behind it? You also mention other choices but do not use them in the experiments, why?
Fourth, why you didn’t try to use a normalized likelihood score? It seems that the unnormalized score is biased towards the shorter sentences. And how do you aggregate the last hidden layer of the encoder to get the sentence representation for the Mahalanobis score?
Fifth, when you compare computation time in Table 4, do you add the time to generate the sequence? It seems that the Mahalanobis distance doesn’t need generated sequence (y) at all, but the proposed methods need it. Thus, the comparison could be unfair in Table 4.

Strengths:
1.    The proposed OOD detection approach shows strong performance and doesn’t degrade the performance of the model on the downstream task, moreover, it could improve it by filtering noisy in-distribution instances.
2.    The method is fully unsupervised, i.e. no OOD examples are needed for it. The method is model and task-independent.
3.    The method is clearly described and the paper is well-written.
Weaknesses:
1.    Lack of testing on other benchmarks on the OOD detection – NMT or classification. See papers [1]-[3].
2.    Lack of support for some of the statements and choices (see Questions)

After rebuttal I increase the soundness scor

**Questions For The Authors:**

A. It is not clear why we choose to filter 80% of in-distribution samples. Do we choose it on some held-out data? Again, how do you choose hyper-parameters for the model if you do not use any OOD examples?

B. Why do we choose the uniform distribution as the reference, and what is the intuition behind it? You also mention other choices but do not use them in the experiments, why?

C. Why you didn’t try to use a normalized likelihood score? It seems that the unnormalized score is biased towards the shorter sentences. And how do you aggregate the last hidden layer of the encoder to get the sentence representation for the Mahalanobis score?

D. When you compare computation time in Table 4, do you add the time to generate the sequence? It seems that the Mahalanobis distance doesn’t need generated sequence (y) at all, but the proposed methods need it. Thus, the comparison could be unfair in Table 4.

**Reasons To Accept:**

1.    The proposed OOD detection approach shows strong performance and doesn’t degrade the performance of the model on the downstream task, moreover, it could improve it by filtering noisy in-distribution instances.
2.    The method is fully unsupervised, i.e. no OOD examples are needed for it. The method is model and task-independent.
3.    The method is clearly described and the paper is well-written.

**Reasons To Reject:**

1.    Lack of testing on other benchmarks on the OOD detection – NMT or classification. See papers [1]-[3].
2.    Lack of support for some of the statements and choices (see Questions)

**Reproducibility:**

4: Could mostly reproduce the results, but there may be some variation because of sample variance or minor variations in their interpretation of the protocol or method.

**Reviewer Confidence:**

4: Quite sure. I tried to check the important points carefully. It's unlikely, though conceivable, that I missed something that should affect my ratings.

**Typos Grammar Style And Presentation Improvements:**

It would be better to move the baselines description into the main part of the paper from the appendix, so, the paper will be self-contained.

Page 1, line 16 – ODD -> OOD

In related work line 206 – in classification tasks, the single-class (ID data) Mahalanobis distance can also show decent performance compare to per-class Mahalanobis distance (see [4], [5], [6]).

---

> ### Author Rebuttal · Authors · 2023-08-28
>
> We thank Reviewer j15V for their thorough review of our work.
>
> **Concerning the construction of the benchmark and choice of setting.** We recognize that we missed works [1-3], and we will add them in the revised version of our work. Nevertheless, we can point out several major differences:
> - In [1] Tim Z. Xiao et al., the authors perform experiments on a single pair of close languages (nld-deu to english). In contrast, our work includes many more pairs and OOD type (we consider topic shifts and close language shifts: afr-nld, nld-deu, por-spa, cat-spa). In addition, it is not clear if this work was ever published.
> - We missed reference [2] by Vazhentsev A. et al., however, this work seems to have been released after the submission deadline of EMNLP, which is the main explanation. They propose ensembling methods that rely on different models to assess the uncertainty of a generation and the OOD nature of the sample. In contrast, our methods are added on top of a single existing model. **They report excellent results when languages are widely different (English vs French in terms of OOD, in Table 16 we report similar results for french vs Spanish for example). However, their results on smaller shifts are not as good around 0.75 (Table 9, Reddit dataset), which is consistent with our results on topic shifts (See Table 17), where we achieve similar or higher AUROCs, with methods that do not require ensembling but only the probability distribution output by a single model.**
>
> - In [3] Guerreiro, Nuno M. et al., the authors focus on natural/normal settings, where the inputs are in distribution and not perturbed by focusing on detecting hallucinations. In contrast, our work focuses specifically on handling OOD samples. Although we could test our methods for detecting hallucinations, we believe it would be a very different task.
>
> **Concerning the lack of comparison of our methods in classification tasks.**
> - We focused on generation as OOD detection in text classifiers has already been explored by many high-performing works (See for example, Colombo et al, Neurips 2022 https://arxiv.org/abs/2211.13527).
> - Our work was prompted by the fact that features-based OOD scores rely on being able to compute a score per-class in classification, which is not possible in text generation, and that this assumption is critical but wrong in text generation (see Appendix A). In classification settings, features-based methods that rely on per-class scores achieve near-perfect results, making our methods irrelevant in classification. Thus, the sole focus is on text generation, which has been significantly less explored in previous work.
>
> **Concerning the other works reviewer j15V pointed out:**
> - [4] Podolskiy, A.V., et al, They focus on OOD detection in classification only and not in generation and conduct a study of the performance of common baselines (the same we used MSP, Likelihood and Mahalanobis distance) in transformers for classification.
> - [5] Yiyou Sun, et al. We were not aware of this work from computer vision. The method they propose could very well be leveraged in NLP but it would be a different work and setting as they rely on having white-box access to the model, whereas we only rely on the soft probability output by the model.
>
> **Questions:**
> - Since we do not allow the use of OOD samples in our setting, we decided to calibrate our OOD detection methods using the training data. We assume that a share of the training is of bad quality and should be excluded. We settled on 80% as a reasonable assumption, but we provide results for a threshold of 99% in Table 23.
> - Using the uniform distribution yields the Negentropy of the distribution (Brillouin, 1953), (https://en.wikipedia.org/wiki/Negentropy). We use it as a measure of uncertainty about the prediction. The negentropy consists of the entropy normalized by the size of the support of the distribution. In the case of text generation, while the vocabulary is very large, the tokens with non-zero probability are rare. We are interested in the shape of the distribution over the non-zero probability tokens and not over the whole vocabulary. We did perform experiments using the TF-IDF distributions, but those did not yield positive results, as reported in Appendix H.3.
> - You are right. We made a mistake in reporting the score, we use the normalized likelihood (the likelihood of the sequence as output as a score from the beam search generation). We will revise the paper to correct this inconsistency.
> - We performed experiments taking the average embedding over the input. We also tried using only the embedding of the first token.
> - We only measured the additional time needed to perform the OOD detection when performing the full generation. So, indeed, one could first extract the encoder embeddings, perform OOD detection using the Mahalanobis distance and then proceed with the generation if the input is deemed “IN distribution”. In that case, using the Mahalanobis distance or other “encoder-only based OOD scores” would be more efficient since it eliminates the generation process. However, doing so does not allow us to assess the quality of the generated sentence as our method does.
>
> We hope our answers have alleviated your concerns and you will consider updating our grade accordingly.

---

### Official Review · Reviewer_fLxa · 2023-08-04

**Soundness:** 3

**Excitement:**

3: Ambivalent: It has merits (e.g., it reports state-of-the-art results, the idea is nice), but there are key weaknesses (e.g., it describes incremental work), and it can significantly benefit from another round of revision. However, I won't object to accepting it if my co-reviewers champion it.

**Paper Topic And Main Contributions:**

The paper presents a new framework called RAINPROOF for detecting out-of-distribution (OOD) samples for text generation models. The goal is to detect when an input is different from what the model saw during training, which can cause the model to generate poor or unsafe outputs. RAINPROOF introduces a new benchmark called LOFTER to evaluate OOD detection methods in more realistic scenarios like close language pairs (e.g. Spanish-Catalan) or domain shifts (e.g. news to medical text).

**Reasons To Accept:**

Strength
- RAINPROOF computes an anomaly score based on the model's token probabilities being different from a reference distribution, indicating the input is OOD. It can work without reference data or use a set of in-distribution examples.
- Experiments show RAINPROOF outperforms baselines like maximum softmax probability and improves metrics like BLEU by removing OOD samples. It also provides interpretable detections.

**Reasons To Reject:**

Weakness
- Overall, with the development of LLMs, NLP systems are becoming more and more robust against distribution shifts, such as mastering different languages, topics, and scenarios. What are the truly realistic and challenging OOD problems given nowadays powerful models? I view the tasks and datasets in LOFTER as traditional ones, which may not be important in this era.
- The experiments on LLMs are few and the results are not satisfying. AUROC remains low and FPR is higher than 0.8
- The performance advantage is marginal on large models according to Table 12, 13, and 14
- Sentences are incomplete. Line 289: Softmax-based detectors such as MSP, Energy or the sequence log-likelihood. Table 7: Missing ``Ours.''

**Reproducibility:**

5: Could easily reproduce the results.

**Reviewer Confidence:**

3: Pretty sure, but there's a chance I missed something. Although I have a good feel for this area in general, I did not carefully check the paper's details, e.g., the math, experimental design, or novelty.

---

> ### Author Rebuttal · Authors · 2023-08-28
>
> We want to thank reviewer fLxa for their review of our work.
>
> Weaknesses:
> - **About the LLM era and relevancy of our work on smaller specialized models**: While we do agree that our results are not satisfying on the most recent LLMs and that our benchmark focuses on traditional NLP tasks and datasets, these larger models cannot be deployed everywhere or by everyone (yet?). Many still rely on smaller specialized architectures. For example, the Marian MT models [1] are such bilingual models optimized for fast translation. In addition, the definition of OOD becomes less clear regarding large language models trained on various tasks and datasets. We would argue that OOD detection especially matters for smaller specialized models intended for specific usage (in our case, translation). Therefore, we believe our work remains relevant if we want to use smaller/more efficient and highly specialized models.
>
> - In addition as shown in [2] OOD samples are still a challenge and an open question for LLM as they significantly impact their performance. Generally speaking, abstention methods are still very much needed and relevant, from smaller specialized models to LLMs. It is also reasonable to believe that the abstention methods for LLMs and smaller models should differ because of the widely different behaviours exhibited by LLMs and smaller models.
>
> - We will correct the typos you pointed out in the camera-ready version of our work.
>
> We hope our answers have alleviated your concerns, and you might be able to update your score accordingly.
>
>
> [1] Junczys-Dowmunt et al., Marian: Fast Neural Machine Translation in C++, ACL 2018
>
> [2] Wang and al, On the Robustness of ChatGPT: An Adversarial and Out-of-distribution Perspective

---

### Official Review · Reviewer_DVtH · 2023-08-04

**Soundness:** 3

**Excitement:**

4: Strong: This paper deepens the understanding of some phenomenon or lowers the barriers to an existing research direction.

**Missing References:**

Some reference on the OOD detection tasks are missing:

Estimating Soft Labels for Out-of-Domain Intent Detection

A recent survey can also be found here:

A Survey on Out-of-Distribution Detection in NLP


**Paper Topic And Main Contributions:**

This paper try to perform OOD detection on NLG tasks. Specifically, they propose to use an everaged generation probability as an OOD score to determine whether a given (x, y) pair is an OOD sample. Detailed analysis and discussions are given in this paper. The authors also claim that their method can improve the final performance of the NLG model.

**Questions For The Authors:**

1. in line 289-290 "Softmax-based detectors such as MSP, Energy or the sequence log-likelihood" does not have a verb.



**Reasons To Accept:**

1. The authors propose an interesting task of detection OOD sample based on the input and output pair (x, y) for NLG model.

2. The analysis and discussions in the paper are comprehensive and solid.

3. The proposed OOD detection method is effective.

**Reasons To Reject:**

1. It is better to detail how each baseline is implemented? For example, for the baseline MSP, how do you calcuate the detection score. It is better to illustrate this with equations. (since you have mentioned that "Due to the large vocabulary size, it is unclear how these methods generalize to sequence generation tasks.")

2. Usually the OOD detection process is performed only on the input x, i.e., if x is an OOD sample, then we should not feed it into the model to produce y. The proposed method need to get access to the input and output pair (x, y). Sometimes, generating the output y for OOD inputs may lead to unpredictable behaviour of downstream systems.

**Reproducibility:**

4: Could mostly reproduce the results, but there may be some variation because of sample variance or minor variations in their interpretation of the protocol or method.

**Reviewer Confidence:**

4: Quite sure. I tried to check the important points carefully. It's unlikely, though conceivable, that I missed something that should affect my ratings.

---

> ### Author Rebuttal · Authors · 2023-08-28
>
> We thank DVtH for their review.
>
> - **Our methods do not require the true output $y$. They rely on the output $\hat y$ generated by the model.** Making the methods perfectly suitable for real-world applications.
> - We give in Appendix B.7 the details on how all the baselines are computed. We take the average OOD score along the generated sequence for MSP and Energy. This is the naive most straightforward method to generalize these baselines. In [1] suggested by reviewer j15V, the authors rely on similar generalizations of the baselines across the elements of the beam.
>
> We will fix the typos you raised to our attention and include the missing references you pointed out.
>
> [1] Vazhentsev A. et al. Efficient Out-of-Domain Detection for Sequence to Sequence Models. in ACL 2023

---

### Meta-Review · Area_Chair_SrFW · 2023-09-25

**Recommendation:** 4

**Metareview:**

This paper addresses the OOD problem in generation (machine translation and response generation in dialogue). The paper propose an evaluation setting, called LOFTER, and show that their proposed method outperforms competitive baselines. In the reviews, There have been concerns about (i) comparison with stronger baselines on established OOD benchmarks on the classification task, (ii) the validity and relevance of the OOD problem in the LLM era, (iii) the marginal improvement of the proposed method in some settings. The authors have addressed these concerns, and by revising the paper according to the discussions, the paper seems to be ready for publication.

---

### Decision · Program_Chairs · 2023-10-07

**Decision:**

Accept-Main

**Comment:**

This paper addresses the OOD problem in generation (machine translation and response generation in dialogue). The paper propose an evaluation setting, called LOFTER, and show that their proposed method outperforms competitive baselines. In the reviews, There have been concerns about (i) comparison with stronger baselines on established OOD benchmarks on the classification task, (ii) the validity and relevance of the OOD problem in the LLM era, (iii) the marginal improvement of the proposed method in some settings. The authors have addressed these concerns, and by revising the paper according to the discussions, the paper seems to be ready for publication.